# Learning Generalizable Agents via Saliency-Guided Features Decorrelation

Sili Huang[1]     Yanchao Sun[2]     Jifeng Hu[3]     Siyuan Guo[4]     Hechang Chen*[5]
Yi Chang[6]*     Lichao Sun[7]     Bo Yang[8]*

[1,8]Key Laboratory of Symbolic Computation and Knowledge Engineering of Ministry of Education
[1,3,4,5,6]School of Artificial Intelligence, Jilin University, China
[2]Department of Computer Science, University of Maryland, College Park, USA
[7]Lehigh University, Bethlehem, Pennsylvania, USA

## Abstract

In visual-based Reinforcement Learning (RL), agents often struggle to generalize well to environmental variations in the state space that were not observed during training. The variations can arise in both task-irrelevant features, such as background noise, and task-relevant features, such as robot configurations, that are related to the optimal decisions. To achieve generalization in both situations, agents are required to accurately understand the impact of changed features on the decisions, i.e., establishing the true associations between changed features and decisions in the policy model. However, due to the inherent correlations among features in the state space, the associations between features and decisions become entangled, making it difficult for the policy to distinguish them. To this end, we propose Saliency-Guided Features Decorrelation (SGFD) to eliminate these correlations through sample reweighting. Concretely, SGFD consists of two core techniques: Random Fourier Functions (RFF) and the saliency map. RFF is utilized to estimate the complex non-linear correlations in high-dimensional images, while the saliency map is designed to identify the changed features. Under the guidance of the saliency map, SGFD employs sample reweighting to minimize the estimated correlations related to changed features, thereby achieving decorrelation in visual RL tasks. Our experimental results demonstrate that SGFD can generalize well on a wide range of test environments and significantly outperforms state-of-the-art methods in handling both task-irrelevant variations and task-relevant variations.

## 1   Introduction

Learning control from high-dimensional visual observations is a critical requirement for real-world applications and has received significant attention in recent years [43, 18, 10]. Reinforcement Learning (RL) has demonstrated its effectiveness in solving visual tasks by learning from compact representations of sensory inputs [12, 13]. However, current algorithms exhibit limited reliability when deployed in unseen environments, despite achieving satisfactory performance during training [7, 48]. It remains an open challenge to learn policies with good generalization across both changed task-irrelevant and relevant features [24]. Consider a simple example during the evaluation stage: a well-trained robot may fail to accomplish the task when confronted with task-irrelevant visual disturbances or changes to the task-relevant robotic arm, as illustrated in Figure 1.

Generalization to task-irrelevant features in vision RL tasks has been widely discussed [49, 11, 6, 3]. Since it is a well-established consensus that task-irrelevant features should not affect the decision, agents are typically designed to maintain an invariant policy when these features are changed

---

*Corresponding Author. Bo Yang, Hechang Chen and Yi Chang.

37th Conference on Neural Information Processing Systems (NeurIPS 2023).

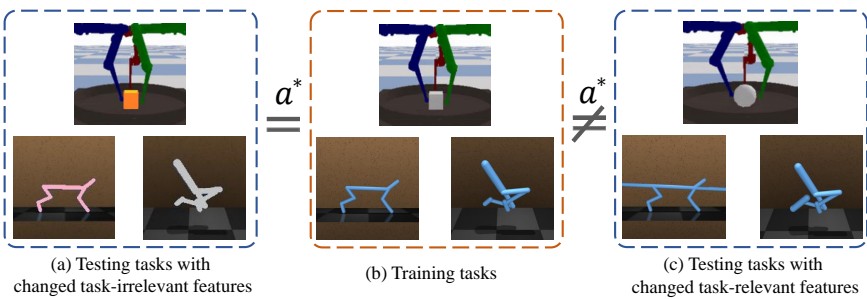

(a) Testing tasks with changed task-irrelevant features

(b) Training tasks

(c) Testing tasks with changed task-relevant features

Figure 1: Visualizing tasks across different variations. $a^*$ denotes the optimal decision, which should be invariant to task-irrelevant features but different to changed task-relevant features. Note that 'changed feature' refers to a feature whose value has changed across tasks.

[25, 28, 31]. To training the invariant policy, previous works have proposed many solutions, such as bisimulation metric [2, 33], data augmentation [46], and contrastive learning [29], to learn invariant representations that filter out the environmental variations. However, the invariant representations may ignore the changed task-relevant features that lead to different optimal decisions, as shown in Figure 1(c). Although Dunion et al. [15] proposed a disentangled representation to preserve task-relevant features, this approach could only recover the sub-optimal decisions in the unseen environment. Therefore, a desired policy should not only remain invariant to task-irrelevant features but also adapt to changed task-relevant features.

Fundamentally, generalization fails because the agent does not understand how the changed features, whether they are task-irrelevant or relevant, affect its decisions. From the perspective of causal inference [14, 35, 39], the associations between features and decisions are often confused with each other, which can be attributed to the inherent correlations among input features in the training data. For instance, in a grasping task, heavier objects may more frequently be placed at lower horizontal heights, which could mislead the robot into adjusting its force based on height. Therefore, a promising approach is to eliminate these correlations, enabling the agent to distinguish the true association between each input feature and decisions [51, 26]. However, recent studies have indicated that achieving perfect decorrelation between all pairs of input features is challenging [45, 44]. This difficulty is especially pronounced in visual RL tasks, as the high-dimensional images that maintain sequential relationships result in complex non-linear and strong correlations among features [40].

To this end, we propose the Saliency-Guided Features Decorrelation (SGFD) method to eliminate the inherent correlations among input features through sample reweighting. Concretely, SGFD consists of two core components designed to address the challenges presented by non-linear and strong correlations in visual RL tasks. The first component is Random Fourier Functions (RFF), which has been demonstrated to assist in approximating the non-linear correlations with linear computational complexity [51]. SGFD employs the cross-covariance matrix in conjunction with RFF to estimate the non-linear correlations in high-dimensional images. To address the strong correlations, SGFD integrates the second component, a classification model specifically designed to distinguish the sources of the images. Given that the values of changed features are unique to each environment, these features can be pinpointed by computing a saliency map [5] of the classification model. Guided by this saliency map, SGFD reweights the training samples to minimize the estimated correlations, with a particular focus on the identified features. To evaluate SGFD, we conduct various experiments on visual RL tasks from DeepMind Control Suite [41] and Causal World [4], which range from scenes with different task-irrelevant features, such as distracting background videos, to task-relevant features, such as the length of robotic arms. The experimental results demonstrate that our SGFD method can significantly improve the generalization across various types of environmental variations.

**Summary of Contributions.** (1) We propose the SGFD model, which achieves improved generalization in visual RL tasks, covering both task-relevant and task-irrelevant situations. (2) We design a sample reweighting method for RL tasks that encourages the agent to understand the impact of changed features on its decisions. (3) We validate the performance of our algorithm through various experiments and demonstrate that SGFD significantly outperforms state-of-the-arts in handling changed task-relevant features.

## 2 Related Work

**Generalization in Image-Based Reinforcement Learning.** Numerous efforts have been made to achieve generalization in image-based RL tasks, employing strategies such as image augmentation [19, 20], encoding inductive biases [3, 29, 36], and learning invariant representations [49, 8], among other approaches [52, 9]. The fundamental idea behind image augmentation is to bolster the robustness of representations by introducing image perturbations [7, 30, 21]. Kostrikov et al. [25] applied an array of augmentation techniques, including cutouts, translation, and cropping, to the RL model. These augmented images can also be utilized to construct an auxiliary task for encoding inductive biases [32]. Laskin et al. [28] sought to learn a representation that maximizes the similarity between different augmentations of the same observation. However, augmentation techniques may not alter the task-relevant features because they can impact the rewards signal [22] and optimal decision-making. Invariance techniques strive to learn a representation that disregards distractors in the image, thereby efficiently generalizing to unseen backgrounds [48, 11, 31]. Recently, there has been an increasing interest in using varying task-relevant situations as an additional experimental setup to evaluate a model's generalization ability [47]. To address this situation, Dunion et al. [15] developed a representation that compresses changed task-relevant features into a small dimension, ensuring that the learned behaviors exhibit minimal changes in the test environment. However, this method requires slight fine-tuning to recover the optimal decisions in the testing environment.

**Features Decorrelation in Generalization.** A promising direction to achieve generalization is through features decorrelation, which eliminates the correlation between inputs features[14, 26]. Shen et al. [37], Kuang et al. [27] attempted to achieve global decorrelation on linear regression tasks by using sample reweighting. Following this, Zhang et al. [51] empirically extended the reweighting method to deep learning models, noting its great potential. Recently, Xu et al. [44] conducted a theoretical analysis of feature decorrelation and proved that perfect reweighting could help select task-relevant features regardless of whether the data generation mechanism is linear or nonlinear. However, it is not easy to eliminate the correlation between all input features under finite training samples[45]. This is especially serious in RL tasks because data often exhibit a sequential relationship [40], resulting in strong correlations between features. To compensate for the imperfection of sample reweighting, Yu et al. [45] introduce a sparsity constraint under the assumption that the correlation between the changed features and other features is significantly smaller. However, this assumption is not applicable to RL tasks because the changed features could be task-relevant and, therefore, significantly correlated with other features in the task. In contrast, we introduce a saliency-guided model to identify the changed features without additional assumptions.

## 3 Preliminaries

**Markov Decision Process.** We define an environment as a Markov Decision Process (MDP) [34], represented by a tuple $\mathcal{M} = (\mathcal{S}, \mathcal{A}, \mathcal{P}, \mathcal{R}, \gamma)$, where $\mathcal{S}$ denotes the high-dimensional state space, $\mathcal{A}$ is the action space, $\mathcal{P}(s'|s, a)$ is the state-transition function, $\mathcal{R} : \mathcal{S} \times \mathcal{A} \rightarrow \mathbb{R}$ is the reward function, and $\gamma \in [0, 1)$ is the discount factor. A decision-making policy, parameterized by $\phi$, is a function $\pi_\phi(a|s)$ that maps a state to distributions over actions. The objective of RL agents is to find a policy that maximizes the expected cumulative return, expressed as $\mathbb{E}_{\pi_\phi}[\sum_{t=0}^{\infty} \gamma^t \mathcal{R}(s_t, a_t)]$.

**Soft Actor-Critic.** We utilize Soft Actor-Critic (SAC) [17] as our base RL algorithm. SAC aims to maximize a variant of the RL objective augmented with an entropy term: $\mathbb{E}[\sum_t \gamma^t \mathcal{R}(s_t, a_t) + \alpha \mathcal{H}(\pi(\cdot|s_t))]$. To optimize this objective, SAC learns two models: a state-action value function $\mathcal{Q}_\theta(s, a)$ and a stochastic policy $\pi_\phi(a|s)$, where $\theta$ and $\phi$ are the parameters for $\mathcal{Q}_\theta(s, a)$ and $\pi_\phi(a|s)$, respectively. The parameters $\theta$ of the state-value function are trained by minimizing the soft Bellman residual, given as:

$$\mathcal{J}_\mathcal{Q}(\theta) = \mathbb{E}_{(s_t, a_t) \sim \mathcal{D}}[\frac{1}{2}(\mathcal{Q}_\theta(s_t, a_t) - (r(s_t, a_t) + \gamma \mathbb{E}_{s_{t+1} \sim \mathcal{P}(\cdot|s_t, a_t)}[V_{\bar{\theta}}(s_{t+1})]))^2], \quad (1)$$

where $\mathcal{D}$ is the replay buffer storing the data, $V_{\bar{\theta}}$ denotes the value function, and $\bar{\theta}$ are the target parameters of $\theta$. The stochastic policy $\pi_\phi(a|s)$ is trained by maximizing the following objective

$$\mathcal{J}_\pi(\phi) = \mathbb{E}_{s_t \sim \mathcal{D}}[\mathbb{E}_{a_t \sim \pi_\phi}[\alpha \log(\pi_\phi(a_t|s_t)) - \mathcal{Q}_\theta(s_t, a_t)]], \quad (2)$$

where $\alpha$ is a temperature coefficient.

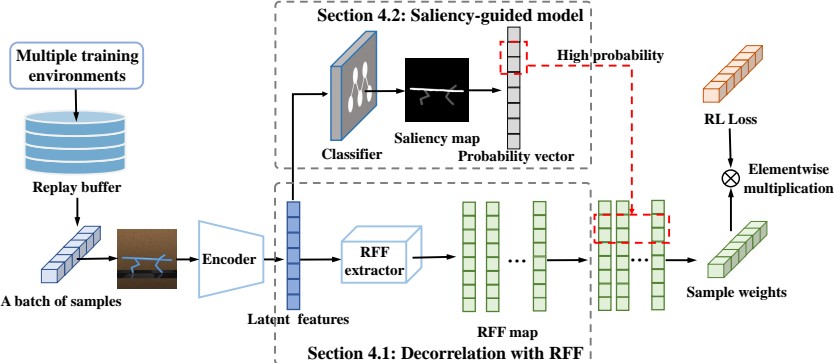

Figure 2: The architecture of our SGFD. SGFD aims to reduce correlations in image features by reweighting samples. This involves five steps: (1) We fetch a sample batch from the replay buffer, which may come from multiple environments with different backgrounds or robot configurations. (2) The image in the sample is compressed by an encoder into latent features. (3) Then, we augment these features using multiple Random Fourier Functions to capture nonlinear correlations. (4) Concurrently, we train a classifier and apply saliency maps to detect features that shift across environments. (5) Finally, SGFD reweights samples to eliminate the correlations between identified features and other features.

**Notations.** For existing visual RL methods [10, 18, 43], the state $s$ is typically compressed into a compact vector representation $\mathbf{z}$. Therefore, we use $\mathbf{z}$ as the default in our calculations unless stated otherwise. For clarity, we define $\mathbf{Z}_j$ as the $j$-th feature in the $d$-dimensional vector, and $\mathbf{z}_{i,j}$ as the value of $\mathbf{Z}_j$ in sample $i$. For instance, if $\mathbf{Z}_j$ corresponds to the angle of a robot arm, $\mathbf{z}_{i,j}$ would represent the encoded value of the angle in image $i$.

# 4 Method

In this section, we introduce the SGFD method, a sample reweighting technique designed to improve generalization across a broad spectrum of visual RL tasks, as depicted in Figure 2. Specifically, SGFD utilizes RFF to enrich image features, thereby enabling the estimation of non-linear correlations in high-dimensional images, as detailed in Section 4.1. Then, SGFD integrates a classifier model and a saliency map to detect changed features across environments. Based on the estimated correlation between identified features and other input features, SGFD adjusts the weights of the samples to achieve decorrelation, as elaborated upon in Section 4.2.

## 4.1 Decorrelation with RFF

To eliminate the correlation between any pair of features, the first step is to quantify their correlation. Given a pair of features $(\mathbf{Z}_i, \mathbf{Z}_j)$ in the RL states, we gather a batch of $n$ samples from the replay buffer $\mathcal{D}$, denoted as $(\mathbf{z}_{1,i}, \mathbf{z}_{1,j}), (\mathbf{z}_{2,i}, \mathbf{z}_{2,j}) \ldots (\mathbf{z}_{n,i}, \mathbf{z}_{n,j})$. The primary challenge lies in accurately estimating the correlation between these two features based on the available samples.

A renowned method for evaluating independence is the Hilbert-Schmidt Independence Criterion (HSIC), which calculates a cross-covariance operator in the Reproducing Kernel Hilbert Space[16]. However, the computational cost of HSIC can be considerable, especially when the batch size $n$ is large. To overcome this issue and apply HSIC to modern RL models, we can compute the independence using the Frobenius norm, shown to be equivalent to the Hilbert-Schmidt norm in Euclidean space [38]. The cross-covariance matrix can be formulated as follows:

$$\hat{\Sigma}_{\mathbf{Z}_i \mathbf{Z}_j} = \frac{1}{n-1} \sum_{k=1}^{n} [(\mathbf{u}(\mathbf{z}_{k,i}) - \mathbb{E}[\mathbf{u}(\mathbf{Z}_i)])^T \cdot (\mathbf{v}(\mathbf{z}_{k,j}) - \mathbb{E}[\mathbf{v}(\mathbf{Z}_j)])], \tag{3}$$

where $\mathbb{E}[\mathbf{u}(\mathbf{Z}_i)] = \frac{1}{n} \sum_{k=1}^{n} \mathbf{u}(\mathbf{z}_{k,i})$, $\mathbf{u}$ and $\mathbf{v}$ are vectors, and each component is a function sampled from the RFF space, formulated as follows:

$$\begin{aligned}
\mathbf{u}(\mathbf{z}_{:,i}) &= (u_1(\mathbf{z}_{:,i}), u_2(\mathbf{z}_{:,i}), \ldots, u_M(\mathbf{z}_{:,i})), u_m(\mathbf{z}_{:,i}) \in \mathcal{H}_{\mathrm{RFF}}, \forall m, \\
\mathbf{v}(\mathbf{z}_{:,j}) &= (v_1(\mathbf{z}_{:,j}), v_2(\mathbf{z}_{:,j}), \ldots, v_M(\mathbf{z}_{:,j})), v_m(\mathbf{z}_{:,j}) \in \mathcal{H}_{\mathrm{RFF}}, \forall m,
\end{aligned} \tag{4}$$

where $\mathcal{H}_{\text{RFF}}$ denotes the RFF space and has been demonstrated to approximate non-linear correlation between image features [51]. Specifically, we sample $M$ functions from the RFF space, defined as $\mathcal{H}_{\text{RFF}} = \{h : x \to \sqrt{2}\cos(\omega x + \psi) | \omega \sim \mathcal{N}(0, 1), \psi \sim \text{Uniform}(0, 2\pi)\}$, where $\omega$ is sampled from a standard normal distribution and $\psi$ is sampled from a uniform distribution. The independence between $\mathbf{Z}_i$ and $\mathbf{Z}_j$ can be calculated by the Frobenius norm of the cross-covariance matrix $||\hat{\Sigma}_{\mathbf{Z}_i\mathbf{Z}_j}||_F^2$. If $||\hat{\Sigma}_{\mathbf{Z}_i\mathbf{Z}_j}||_F^2$ is zero, the two features $\mathbf{Z}_i$ and $\mathbf{Z}_j$ are considered independent.

We reformulate the task of correlation elimination as an optimization problem aimed at minimizing the Frobenius norm of the cross-covariance matrix $||\hat{\Sigma}_{\mathbf{Z}_i\mathbf{Z}_j}||_F^2$. To this end, we introduce a set of learnable sample weights, denoted by $\mathbf{w} \in \mathbb{R}$, subject to the constraint that $\sum_{k=1}^{n} w_k = n$. Based on the weighted samples, the independence can be defined as:

$$\hat{\Sigma}_{\mathbf{Z}_i\mathbf{Z}_j;\mathbf{w}} = \frac{1}{n-1} \sum_{k=1}^{n} [(w_k \mathbf{u}(\mathbf{z}_{k,i}) - \mathbb{E}[\mathbf{w}\mathbf{u}(\mathbf{Z}_i)])^T \cdot (w_k \mathbf{v}(\mathbf{z}_{k,j}) - \mathbb{E}[\mathbf{w}\mathbf{v}(\mathbf{Z}_j)])], \quad (5)$$

where $\mathbb{E}[\mathbf{w}\mathbf{u}(\mathbf{Z}_i)] = \frac{1}{n} \sum_{k=1}^{n} w_k \mathbf{u}(\mathbf{z}_{k,i})$. The correlation among all features in RL states can be eliminated by solving the following optimization problem:

$$\mathbf{w}^* = \arg\min_{\mathbf{w}} \sum_{1 \leq i < j \leq d} ||\hat{\Sigma}_{\mathbf{Z}_i\mathbf{Z}_j;\mathbf{w}}||_F^2, \quad (6)$$

where $d$ is the dimension of $\mathbf{Z}$. When the number of samples is not limited, Equation (6) is proved to have multiple solutions that achieve perfect decorrelation [27].

## 4.2 Saliency-Guided Decorrelation with RFF

In practice, achieving desirable weights that ensure perfect independence between all pairs of features with finite samples is a significant challenge [45]. Therefore, relying solely on the features decorrelation approach introduced in Section 4.1 may not be sufficient. Furthermore, in RL tasks, the sequential nature of the data creates strong dependencies between features, further complicating the sample reweighting. To overcome this challenge, we propose a saliency-guided model that prioritizes the most crucial correlations between changed features and other image features.

The saliency map [5], a well-known computer vision method, uses the backpropagation algorithm to compute the attributions on neural networks. It is widely used to interpret models by visualizing which pixels are more important for decision-making. Inspired by the work of Bertoin et al. [7], we incorporate this functionality into the training phase by introducing a classifier model that distinguishes the environmental source of states. The contribution of each feature can be determined by computing a saliency map. For example, the contribution of the $i$-th feature in the state is $M_i(f, Z) = \partial f(\mathbf{Z})/\partial \mathbf{Z_i}$, where $f$ denotes the classifier. If the $i$-th feature changes across the environments, it will provide critical information for the classifier's decisions, leading to a large value of $M_i(f, Z)$.

Given our focus on the correlations related to changed features, it is anticipated that when a pair of features is more likely to be part of the changed set, they should be prioritized for decorrelation. Accordingly, the probability that each feature belongs to the changed part can be calculated by the saliency map, i.e., $p(Z_i) = e^{M_i} / \sum_{1 \leq j \leq d} e^{M_j}$. We incorporate this term into the Equation (6), revising it as:

$$\mathbf{w}^* = \arg\min_{\mathbf{w}} \sum_{1 \leq i < j \leq d} p(\mathbf{Z}_i)p(\mathbf{Z}_j)||\hat{\Sigma}_{\mathbf{Z}_i\mathbf{Z}_j;\mathbf{w}}||_F^2. \quad (7)$$

Based on the weighted samples, we can reformulate the Equation (2) as:

$$\mathcal{J}_\pi(\phi) = \sum_{s_t \sim \mathcal{D}} w_t \mathbb{E}_{a_t \sim \pi_\phi}[\alpha \log(\pi_\phi(a_t|s_t)) - \mathcal{Q}_\theta(s_t, a_t)]. \quad (8)$$

With the reformulated Equation 8, in which the changed features are considered independently from other features, the agent can learn the actual association with output decisions. More details of SGFD are described in Appendix A and Appendix B.

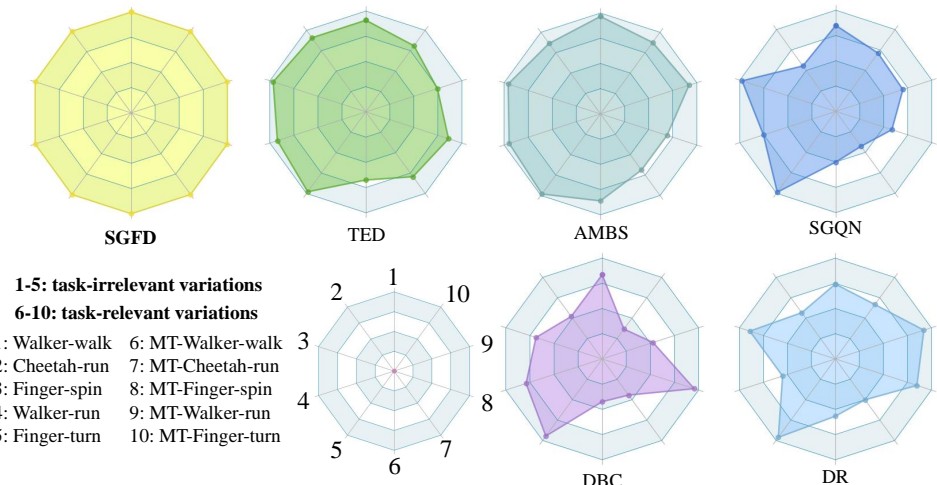

Figure 3: The normalized performance of generalization under changed task-irrelevant and relevant features. Each polygon represents one algorithm across 10 tasks. Each vertex of the polygon denotes the normalized performance, which matches counterclockwise from walker-walk to MT-finger-turn. Note that the results related to task-relevant features are the average of the two settings presented in Table 2.

## 5 Experiments

To carry out a comprehensive investigation of SGFD, we have performed a series of experiments on visual RL tasks, utilizing the DeepMind Control Suite [41] and Causal World [4]. We provide a brief introduction to the tasks, baselines, and evaluation metrics in Section 5.1. In Section 5.2, we verify the generalization capabilities of SGFD across a variety of tasks, ranging from scenarios with diverse task-irrelevant features to those with task-relevant features. To gain deeper insight into the workings of SGFD, we evaluated its decorrelation potential and visualized the weighted samples, as detailed in Section 5.3. Then, we conducted several ablation experiments on RFF and saliency-guided model in Section 5.4. Finally, we investigated the interpretability of RL models in Appendix C.

### 5.1 Experiment Setting

**Environments.** The DeepMind Control Suite [41] is a widely-used visual control toolkit that offers a wealth of simulated animal behavior tasks. Drawing on previous works [48, 50], we constructed environments with both background noises and varying robot settings to test the generalization capabilities of the SGFD. The backgrounds were sampled from the Kinetics dataset [23], while the robot settings followed those proposed by Zhang et al. [50]. Furthermore, we utilized the Causal World [4], which comprises a series of robot manipulation tasks. In these manipulation tasks, we employed perception states to evaluate the potential of SGFD for decorrelation. More implement details are provided in Appendix B.

**Baselines.** We compared our method against several recently established baselines, which we outline as follows. Our baseline for disentanglement representation is TED [15]. TED compresses changed features into a small dimension, thereby ensuring that learned behaviors exhibit a minimal change in the testing environment. We used DBC [48] as a baseline method that learns invariant representations based on the bisimulation metric. We also compared with AMBS [11], an enhanced version of DBC that integrates a contrastive method. In addition, we compared with SGQN [7], a novel method that uses data augmentation and the saliency map to ignore the task-irrelevant features in the images. Lastly, we included Domain Randomisation (DR) [42] in our comparison, a technique commonly utilized in control tasks.

**Evaluation Metrics.** The primary metric under consideration is the cumulative reward from the testing environments, which reflects the generalization capabilities of the learned policy models. Additionally, we employed the Pearson correlation coefficient to assess the correlation based on samples, both with and without weighting. Lastly, we utilized the saliency map as an interpretability metric to visualize the learned policy models. All experimental assessments calculate the mean and standard deviation of the results across five seeds unless otherwise stated.

Table 1: Generalization to changed task-irrelevant features. For each task, we cost 1e6 steps in training environments and evaluate in the testing environment under background noises.

| Tasks | SGFD | TED | AMBS | SGQN | DBC | DR |
|---|---|---|---|---|---|---|
| walker-walk | **959.1**$\pm$ **26.3** | 871.5$\pm$ 60.6 | 926.7$\pm$ 53.2 | 815.1$\pm$ 53.9 | 800.9$\pm$ 41.4 | 712.4$\pm$ 93.7 |
| cheetah-run | **599.6**$\pm$ **47.2** | 544.7$\pm$ 22.9 | 517.7$\pm$ 73.4 | 332.8$\pm$ 55.1 | 312.1$\pm$ 20.3 | 340.0$\pm$ 44.0 |
| finger-spin | **965.7**$\pm$ **45.9** | 932.1$\pm$ 71.0 | 925.1$\pm$ 50.5 | 943.3$\pm$ 46.2 | 663.7$\pm$ 68.7 | 860.8$\pm$ 42.1 |
| walker-run | **420.7**$\pm$ **39.2** | 387.8$\pm$ 27.1 | 398.7$\pm$ 32.0 | 317.2$\pm$ 34.5 | 332.4$\pm$ 37.1 | 231.3$\pm$ 8.9 |
| finger-turn | **984.3**$\pm$ **11.5** | 963.9$\pm$ 94.5 | 966.7$\pm$ 37.0 | 971.3$\pm$ 26.0 | 931.2$\pm$ 41.6 | 947.2$\pm$ 21.7 |
| Total | **3929.5** | 3700.0 | 3734.9 | 3379.7 | 3040.3 | 3091.7 |

Table 2: Generalization to changed task-relevant features. For each task, we cost 1e6 steps in training environments and evaluate in the testing environment with different robot parameters. We consider two setups for evaluation: an interpolation setup and an extrapolation setup where the variations in the task-relevant features are interpolations and extrapolations between the variations of the training environment, respectively.

| | Tasks | SGFD | TED | AMBS | SGQN | DBC | DR |
|---|---|---|---|---|---|---|---|
| Interpolation | MT-walker-walk | **549.4**$\pm$ **42.5** | 471.9$\pm$ 18.3 | 532.5$\pm$ 81.7 | 287.1$\pm$ 34.5 | 245.7$\pm$ 47.4 | 343.8$\pm$ 70.6 |
| | MT-cheetah-run | **395.9**$\pm$ **35.2** | 367.8$\pm$ 42.2 | 298.6$\pm$ 53.5 | 225.7$\pm$ 40.9 | 191.7$\pm$ 23.5 | 216.3$\pm$ 33.1 |
| | MT-finger-spin | **234.1**$\pm$ **11.9** | 201.7$\pm$ 17.9 | 161.3$\pm$ 17.3 | 135.7$\pm$ 16.9 | 221.6$\pm$ 21.5 | 207.1$\pm$ 28.2 |
| | MT-walker-run | **170.3**$\pm$ **05.7** | 125.6$\pm$ 13.2 | 161.4$\pm$ 06.7 | 126.3$\pm$ 18.6 | 97.2$\pm$ 19.0 | 160.3$\pm$ 16.6 |
| | MT-finger-turn | **923.7**$\pm$ **26.1** | 748.9$\pm$ 44.3 | 821.8$\pm$ 52.3 | 786.6$\pm$ 65.6 | 358.5$\pm$ 83.9 | 704.7$\pm$ 70.1 |
| | Total | **2273.4** | 1910.6 | 1975.6 | 1561.4 | 1114.7 | 1632.2 |
| Extrapolation | MT-walker-walk | **541.7**$\pm$ **65.4** | 365.9$\pm$ 17.7 | 467.5$\pm$ 91.7 | 271.2$\pm$ 75.4 | 229.8$\pm$ 89.9 | 307.8$\pm$ 58.9 |
| | MT-cheetah-run | **392.3**$\pm$ **32.1** | 311.9$\pm$ 52.7 | 270.2$\pm$ 35.5 | 167.2$\pm$ 39.1 | 174.0$\pm$ 45.1 | 196.6$\pm$ 49.8 |
| | MT-finger-spin | **231.8**$\pm$ **11.5** | 199.7$\pm$ 18.0 | 160.2$\pm$ 17.6 | 135.6$\pm$ 11.3 | 221.4$\pm$ 43.0 | 197.1$\pm$ 21.5 |
| | MT-walker-run | **170.0**$\pm$ **07.2** | 126.7$\pm$ 13.2 | 156.2$\pm$ 07.5 | 118.9$\pm$ 18.2 | 89.7$\pm$ 19.7 | 156.9$\pm$ 12.7 |
| | MT-finger-turn | **917.3**$\pm$ **22.6** | 743.6$\pm$ 58.3 | 803.5$\pm$ 57.4 | 653.3$\pm$ 56.6 | 335.6$\pm$ 56.5 | 611.7$\pm$ 53.6 |
| | Total | **2253.1** | 1747.8 | 1857.6 | 1346.2 | 1050.5 | 1470.1 |

## 5.2 Evaluation on the Generalization of SGFD

**Evaluation on Changed Task-Irrelevant Features.** In this experiment, we assessed the generalization capability of our SGFD model with respect to different task-irrelevant features. During the training phase, the policy model was allowed to interact with the environments 1 million times. During the evaluation, we assessed the cumulative rewards in a testing environment with previously unseen backgrounds. As indicated in Table 1 and vertices 1−5 of the polygons in Figure 3, our model outperformed other baselines under the background noises. Although AMBS and SGQN aimed to direct the policy model to focus on task-relevant features, the learned representations may still contain partial background features that were not effectively filtered out. Consequently, the policy model established associations with these tasks-irrelevant features, leading to incorrect decision-making when the background changed. In contrast, SGFD enabled the policy to understand that background features should not affect decision-making, thereby achieving superior generalization performance. The results indicate that decorrelation is effective in RL tasks with background noises and that SGFD exhibits strong generalization capabilities to new values of task-irrelevant features.

**Evaluation on Changed Task-Relevant Features** To evaluate the generalization ability of SGFD with respect to task-relevant features, we conducted evaluations under two experimental setups: interpolation and extrapolation. In the interpolation setup, task-relevant features underwent changes that were interpolated between those in the training environments. In contrast, in the extrapolation setup, changes exceeded the range observed in the training environments. As depicted in Table 2 and at vertices 6−10 of the polygons in Figure 3, SGFD demonstrated superior performance under both setups. Generally, extrapolation poses a more significant challenge for generalization, as it necessitates the model to comprehend how changed features impact optimal decision-making. Owing to the associations between changed features and decisions were not confused, our model outperformed other baselines, with more substantial performance gaps observed in the extrapolation

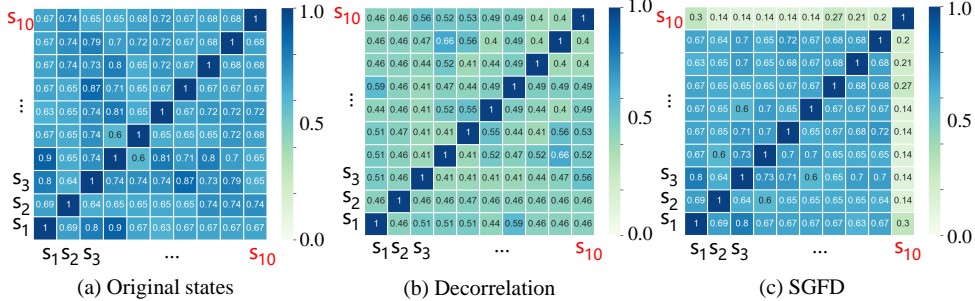

|     | (a) Original states | (b) Decorrelation | (c) SGFD |
|-----|--------------------|--------------------|----------|

Figure 4: Pearson correlation coefficients among features: a) on raw data, b) on weighted data, c) on saliency-guided weighted data. The features $s_1$ to $s_{10}$ represent the state information of the environment and act as conditional inputs for decision-making $\pi_\phi(a|s)$, such as the coordinates of the object or the angle of the manipulator. The feature $s_{10}$ is the one that varies across the environments; thus, the correlations between $s_{10}$ and other features play a crucial role in generalizations. The strength of the correlation is reflected by the color intensity - the darker the color, the stronger the correlation.

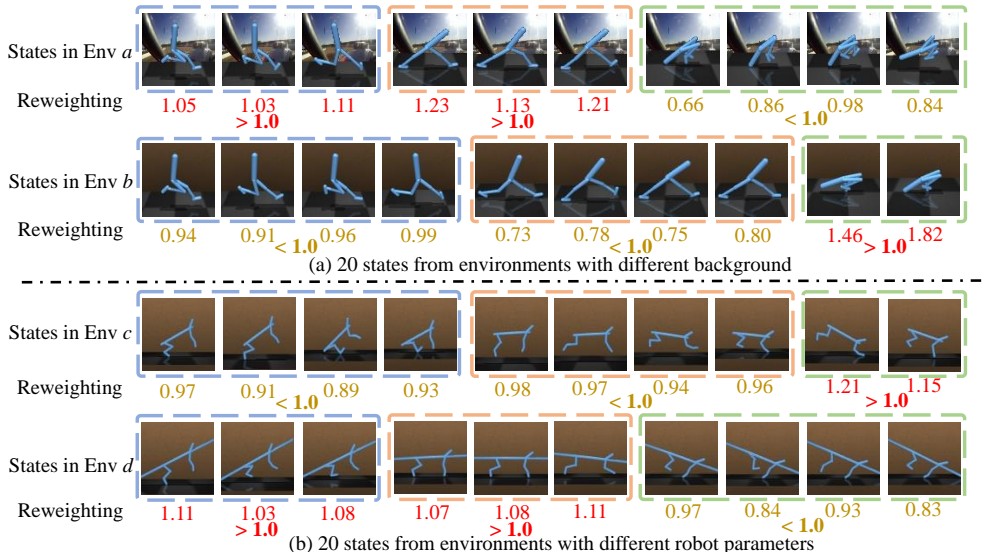

Figure 5: Visualization of weighted samples obtained from different environments with background noises and robot parameters. States that belong to the same dotted box with a specific color are considered semantically similar. SGFD balances the weights of samples to eliminate the correlations between changed features and other features. As shown in (a), due to the interference of the background noises, the robot fell more often (states in the green dotted box) than it did without the background. SGFD reduces the weighting of states with backgrounds and enhances the weighting of states without backgrounds, effectively neutralizing the correlation between background and body posture. A comparable result was observed in (b), where the reweighting process aimed to equalize the number of states in each group across environments with different body lengths.

setting. These results demonstrate that SGFD can effectively adapt to new values of task-relevant features, showing its robust generalization capability in a variety of scenarios.

## 5.3 Case Studies on the SGFD

**Correlation of the Weighted Samples.** To evaluate the decorrelation capability of our model, we implemented a 'Picking' task in Causal World. This task involves a robotic arm grasping objects, enabling the model to access the state with physical meaning. We recorded the Pearson correlation coefficients among features under three conditions: a) on raw data, b) on weighted data, and c) on saliency-guided weighted data. Among these features, we singled out $s_{10}$, a feature that represents mass, as the focal feature for environmental variation. Consequently, when the model is given the task of handling an object with a new mass, it becomes critical to carefully examine the correlations between the mass feature and other features. As depicted in Figure 4, due to limited samples and strong correlations, the effectiveness of direct decorrelation of all features is significantly diminished.

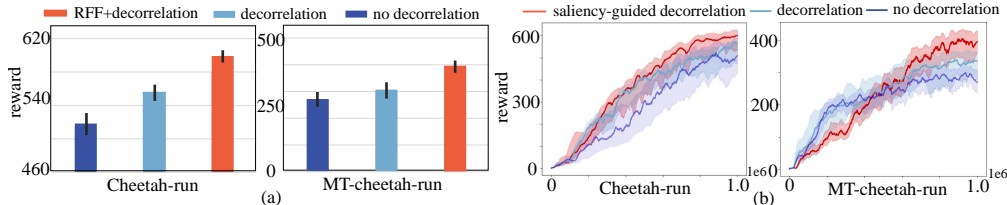

Figure 6: Ablation studies. (a) The performance difference of ablation studies on the RFF. (b) The performance difference of ablation studies on the saliency-guided model. The x-axis denotes the steps in the training environments. The y-axis denotes the cumulative reward recorded in the testing environment with background noises (Cheetah-run) or varied robot parameters (MT-cheetah-run).

In contrast, our SGFD model can effectively reduce the correlations between the mass feature ($s_{10}$) and other features. We also conducted an experiment involving changes in multiple features and obtained similar results. Detailed information about this experiment is described in Appendix C.1.

**Visualization of the Weighted Samples.** To delve deeper into the implications of weighted samples from our SGFD method, we collected 20 state instances during training, each exhibits distinct backgrounds or robot parameters from various environments. The identified states were sorted into three groups in each environment, each consisting of multiple similar state instances. As illustrated in Figure 5 (a), the green dotted boxes indicate several states with "fall" posture. Due to the impact of background noises on the policy, we identified 4 similar states in the environment $a$ and 2 similar states in the environment $b$. After the reweighting process, both environments $a$ and $b$ had approximately 3 similar states, effectively neutralizing the correlation between background and body posture. A similar outcome was observed in Figure 5 (b), wherein the reweighting process endeavored to balance the number of states in each group across environments with varied body lengths. As a result of the correlation removal, the policy model was better equipped to accurately capture the association between the changed features and decision-making.

### 5.4 Ablation Studies

**Ablation on RFF.** To examine the role of RFF, we compared the complete SGFD algorithm, the decorrelation method without RFF, and the algorithm without any decorrelation. The testing environments comprise varied backgrounds and robot parameters. As depicted in Figure 6 (a), the decorrelation without RFF outperforms the approach without decorrelation but is significantly inferior to SGFD. This discrepancy arises due to the complex nonlinear correlations of features within the image, which RFF can approximate. Without RFF, the basic decorrelation method can only eliminate linear correlations, thus achieving limited improvement.

**Ablation on Saliency-Guided Model.** We conduct a separate ablation study on the saliency-guided model to evaluate its impact on the generalization capability of the overall model. Analogous to the RFF study, we test the performance of three distinct approaches: no decorrelation, decorrelation, and saliency-guided decorrelation. As shown in Figure 6 (b), the performance significantly deteriorates when the saliency-guided model is removed from SGFD. This outcome is predictable because directly decorrelating all pairs of features is challenging, thereby complicating the model's ability to discern the associations between the variant features and decisions.

We conduct additional ablation studies for further insights into the roles of RFF and the saliency-guided model. Detailed analyses and results of these studies are available in Appendix C.2. Moreover, we report on the learning curve of the classifier, as shown in Appendix C.3.

## 6 Conclusion, Limitations, and Broader Impact

In this work, we introduce SGFD, a sample reweighting method designed to enhance generalization in visual reinforcement learning tasks across environments with unseen task-irrelevant and task-relevant features. SGFD is composed of two core components: RFF and a saliency-guided model, which empower the RL agent to understand the impact of changed features on its decisions. We assess SGFD using the DMControl benchmark, where it demonstrates significant improvements in generalization across various environmental variations. Furthermore, our case studies conducted on the Causal

World benchmark, as well as the visualization of the weighting, highlight the superior decorrelation achieved by our model. The results from SGFD underscore the potential value of sample reweighting in generalization RL tasks.

In terms of limitations, we note that our work, like many others [48, 11, 15], relies on stacking consecutive frames to approximate a fully observable condition. This might not be optimal for all scenarios, particularly when the changed features cannot be directly observed. A potential solution is to train an inference model that uses a small amount of data from the deployment environment to rapidly predict the potential variations. Regarding societal impacts, we do not anticipate any negative consequences stemming from the practical application of our method.

## Acknowledgments

This work was supported by the National Key R&D Program of China under Grant Nos. 2021ZD0112501 and 2021ZD0112502; the National Natural Science Foundation of China under Grant Nos. U19A2065, U22A2098, 61976102, 62172185, 62206105 and 62202200; the International Cooperation Project of Jilin Province under Grant Nos.20220402009GH.

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

# Appendix

## A Pseudocode of Saliency-Guided Features Decorrelation

In Algorithm 1, we introduce the training process of SGFD. At each step, we apply the policy model $\pi_\phi$ to interact with training environments and sample a batch of transitions along with environment labels, as described in lines 3-5. Before the sample reweighting, we assess the current classifier's accuracy. If the accuracy exceeds 0.9, we proceed with sample reweighting, as described in lines 6-9. In contrast, if the accuracy is lower and $t > 1e5$, we update the classifier based on the cross-entropy loss, as described in lines 9-13. In experiments, the classifier could be quickly converged. Based on the converged classifier, SGFD uses Equation (7) to learn a set of sample weights, as described in lines 14-17. This process also converged quickly, as only 128 parameters (the batch size) needed to be updated. Finally, SGFD uses Equation (1) and Equation (8) to update state-action value function $\mathcal{Q}_\theta$ and policy model $\pi_\phi$ based on the weighted data, as described in lines 18-20.

---

**Algorithm 1** Saliency-Guided Features Decorrelation

---

1: **Input:** Initialize a policy model $\pi_\phi$, a state-action value function $\mathcal{Q}_\theta$, a classifier model $f_\psi$, and a set of training environments with environment labels $\{e^1, e^2, \ldots, e^K\}$, where $e^k$ is a $K$-dimensional one-hot vector and the $k-$th component is 1
2: **Output:** The learnt policy $\pi_{\phi^*}$
3: **for** $t = 1$ **to** $T$ **do**
4:     Applying the policy model $\pi_\phi$ to interact with training environments
5:     Sampling a batch of transitions along with environment labels $(s_i, a_i, r_i, s'_i, e^k_i) \sim \mathcal{D}$
6:     **for** $iter = 1$ **to** $10$ **do**
7:         **if** $t < 1e5$ **or** the accuracy of $f(s) > 0.9$ **then**
8:             Break this cycle
9:         **else**
10:             Sampling a batch of transitions along with environment labels $(s_i, a_i, r_i, s'_i, e^k_i) \sim \mathcal{D}$
11:             Update the classifier model $f_\psi$ by minimizing the following loss
            $\mathcal{L}(\psi) = -\frac{1}{N} \sum_{i=1}^{N} \sum_{j=1}^{K} e^k_{i,j} \log(f_\psi(s_i)_j)$,
            where $e^k_{i,j}$ denotes the $j-$th component of vector $e^k_i$, $f_\psi(s_i)_j$ denotes the predicted probability of the $j-$th class
12:         **end if**
13:     **end for**
14:     Initialize a set of sample weights that can be learned $\mathbf{w}$
15:     **for** $iter = 1$ **to** $10$ **do**
16:         Update the sample weights by minimizing Equation (7)
17:     **end for**
18:     Update the state-action value function $\mathcal{Q}_\theta$ by minimizing Equation (1)
19:     Update $\pi_\phi$ by minimizing Equation (8)
20: **end for**

---

## B Implement Details

**Experimental Setting Details.** For task-irrelevant variations, we follow the prior work [30] to sample several images from Kinetics [23]. In training, we create four parallel environments with different image backgrounds. In testing, we create a new environment with an unseen image background. For task-relevant variations, we follow the prior work [50] to revise the robot configures. The controllable configuration contains 10 different settings for each task, e.g., the length of the robot torso. In training, we create four parallel environments with different robot configures, where the values of configures are sampled from a predefined distribution, i.e., a uniform distribution between 1 and 5. In testing, we exhibit evaluations under two setups: interpolation and extrapolation. In the interpolation setup, task-relevant features changed in a manner interpolated between those in the training environments, whereas in the extrapolation setup, changes exceeded the range of those in the training environments, i.e., the tenth setting with the farthest training distribution. The visualizations are illustrated in Figure 7.

Table 3: Architectures of SGFD.

|  | Hyperparameter | Value |
|---|---|---|
| Encoder architecture | Convolutional layers | 4 |
|  | Latent representation dimension | 50 |
|  | Image size | (84,84) |
|  | Stacked frames | 3 |
|  | kernel size | $3 \times 3$ |
|  | Channels | 3 |
|  | stride | 2 for the first layer, 4 otherswise |
|  | Activation | ReLU |
| Policy and value function | MLP layers | 2 |
|  | Hidden dimension | 1024 |
|  | Activation | ReLU |
| Classifier architecture | MLP layers | 2 |
|  | Hidden dimension | 128 |
|  | Activation | ReLU |

Table 4: Hyperparameters of SGFD.

|  | Hyperparameter | Value |
|---|---|---|
| Training for policy | Optimizer | Adam |
|  | Learning rate | 1e-3 |
|  | Action repeat | 2 |
|  | Replay buffer capacity | 1000000 |
|  | Batch size | 128 |
|  | Target soft-update rate ($\tau$) | 0.01 |
|  | Actor update frequency | 2 |
| Training for sample reweighting | Optimizer | SGD |
|  | momentum | 0.9 |
|  | Learning rate | 1e-2 |
|  | weight decay | 1e-4 |
|  | The number of RFFs $m$ | 5 |

For each experiment, we report the mean and standard deviation over 10 random seeds.

**Compute.** Experiments are carried out on NVIDIA GeForce RTX 3090 GPUs and NVIDIA A10 GPUs. Besides, the CPU type is Intel(R) Xeon(R) Gold 6230 CPU @ 2.10GHz. Each run of the experiments spanned about 12-24 hours, depending on the algorithm and the complexity of the task.

**Architectures.** Following AMBS [11], we use the same encoder architecture, which consists of 4 convolutional layers. The encoder weights are shared between the policy and the state-action value function. Each convolutional layer has a $3 \times 3$ kernel size and 32 channels. The first layer has a stride of 2, and all other layer has a stride of 1. There is a ReLU activation between each convolutional layer. The convolutional layers are followed by a trunk network with a linear layer, layer normalization, and finally, a tanh activation. As we focus on the sample reweighting, the encoder is updated by referring to the AMBS method.

The policy $\pi_\phi$ and the state-action value function $\mathcal{Q}_\theta$ are both 2-layer MLPs with a hidden dimension of 1024. We apply ReLU activations after each layer except the last layer. The classifier is implemented with 2-layer MLPs with a hidden dimension of 128. We also apply ReLU activations after each layer except the last layer. Table 3 shows the values of the architectures for the encoder, the policy, the value function, and the classifier.

**Hyperparameters.** Table 4 shows the hyperparameters used in our SGFD model.

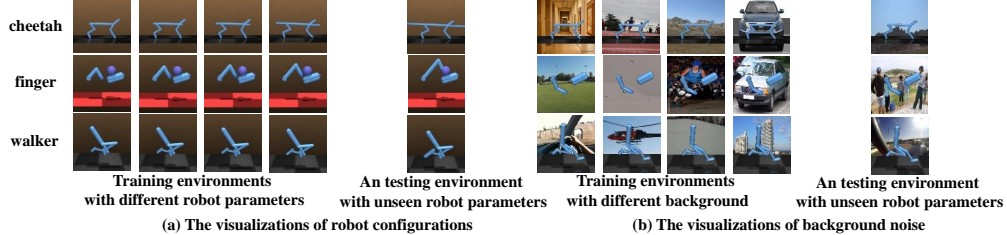

|  | Training environments with different robot parameters | An testing environment with unseen robot parameters | Training environments with different background | An testing environment with unseen robot parameters |
|---|---|---|---|---|

(a) The visualizations of robot configurations    (b) The visualizations of background noise

Figure 7: The visualization of training environments and testing environments.

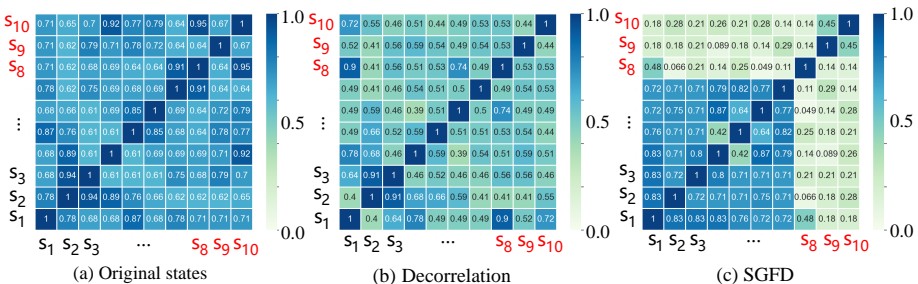

(a) Original states    (b) Decorrelation    (c) SGFD

Figure 8: Pearson correlation coefficients among variables: a) on raw data, b) on weighted data, and c) on saliency-guided weighted data. The variables $S_8$, $S_9$, and $S_{10}$ are the ones that have been changed among the environments.

## C    Additional Experimental Results

### C.1    Additional Correlation Studies on the Weighted Samples

To further evaluate the decorrelation capability of our model, we implement the 'Pushing' task in Causal World and allow the model to access state features that have physical meaning. We record the Pearson correlation coefficients among features under three conditions: a) on raw data, b) on weighted data, and c) on saliency-guided weighted data (SGFD). Among these features, we designate the $S_8$, $S_9$, and $S_{10}$, which represent the length, width, and height, as the features of interest for environmental variations. Therefore, when the model is required to push an object with a new size, the correlation between size and other features needs to be carefully considered. As shown in Figure 8, due to limited samples and strong correlation between features, the effectiveness of directly decorrelating all features is greatly reduced. In contrast, SGFD is capable of effectively reducing the correlation between the size features ($S_8$, $S_9$, and $S_{10}$) and other features.

### C.2    Additional Ablation Studies

We performed additional ablation experiments to test the effect of RFF and the saliency-guided model. As depicted in Figure 9 (a) and Figure 9 (b), when RFF is removed, the generalization performance drops significantly. In contrast, the global decorrelation can cause over-adjusting of samples weighting and impair the performance, as demonstrated by the reward curves in Figure 9 (c). Due to the need to eliminate correlations between variables using limited samples, the model may assign very low weights to certain training samples. However, this can potentially harm the efficiency of training. Thanks to the saliency-guided model, SGFD balances both efficiency and decorrelation.

### C.3    Evaluation on the Classification Model

We recorded the converge curve of the classifier during training. The classifier starts to update when the replay buffer collects $1e5$ steps. To avoid overfitting, we stopped to update the classifier when the accuracy exceeded 0.9. As shown in Figure 10, the classifier converged quickly and achieved the accuracy of 0.9, which means that the classifier does not take up many computing resources.

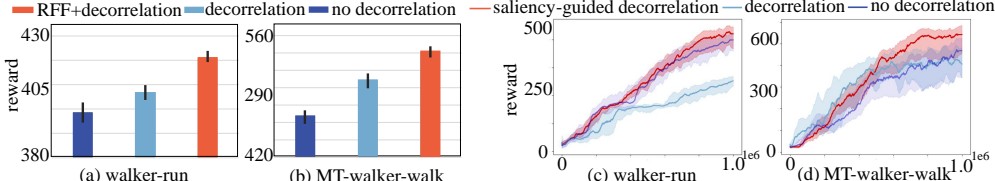

Figure 9: Ablation studies. (a)-(b) The performance difference of ablation studies on the RFF. (c)-(d) The performance difference of ablation studies on the saliency-guided model. The x-axis denotes the steps in the training environments. The y-axis denotes the cumulative reward recorded in the testing environment with background noises (walker-run) or varied robot parameters (MT-walker-walk).

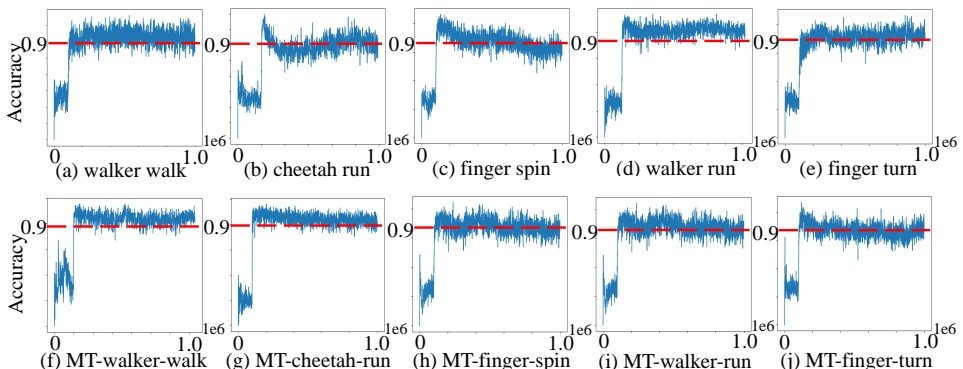

Figure 10: The accuracy of the classifier during training. The x-axis denotes the number of training steps taken in training environments, while the y-axis indicates the accuracy. The classifier starts to update when the replay buffer collects 1e5 steps. The red line corresponds to the 0.9 threshold, which indicates that the classifier stops updating if its accuracy exceeds 0.9.

## C.4    Saliency Maps on the Policy Models

An intuitive approach to explaining visually-based models involves identifying pixels that have a significant impact on the final decision [1]. To determine whether the model focuses on the object or the background during decisions, we visualize the gradient of the action with respect to the input pixels. Experiments are conducted in environments with varying backgrounds and robot parameters, as depicted in Figure 11. Saliency maps of the baseline models reveal that various backgrounds draw considerable attention from the policy model while failing to make decisive contributions to our model. Moreover, SGFD can notice the changing aspects of the robot's body that other baseline models overlook. As a result, the saliency maps demonstrate that the decorrelation benefits the model's generalization in both task-relevant and task-irrelevant situations.

## C.5    Additional Visualization of the Weighted Samples

In this experiment, we significantly expanded the dataset to include 300 state instances, which is substantially larger than the 20 states mentioned in Section 5.3. The identified states were also sorted into three groups in each environment, each consisting of multiple similar state instances. Moreover, we carefully analyzed the distribution of these states within each group, and the corresponding proportions are visually depicted in Figure 12. As illustrated in Figure 12 (a), the green dotted boxes indicate that 50% of the states in environment A and 35.4% of the states in environment B exhibited similarities. After the reweighting process, both environments A and B had approximately 42% similar states, effectively neutralizing the correlation between background and body posture. A similar outcome was observed in Figure 12 (b), wherein the reweighting process endeavored to balance the number of states in each group across environments with varied body lengths.

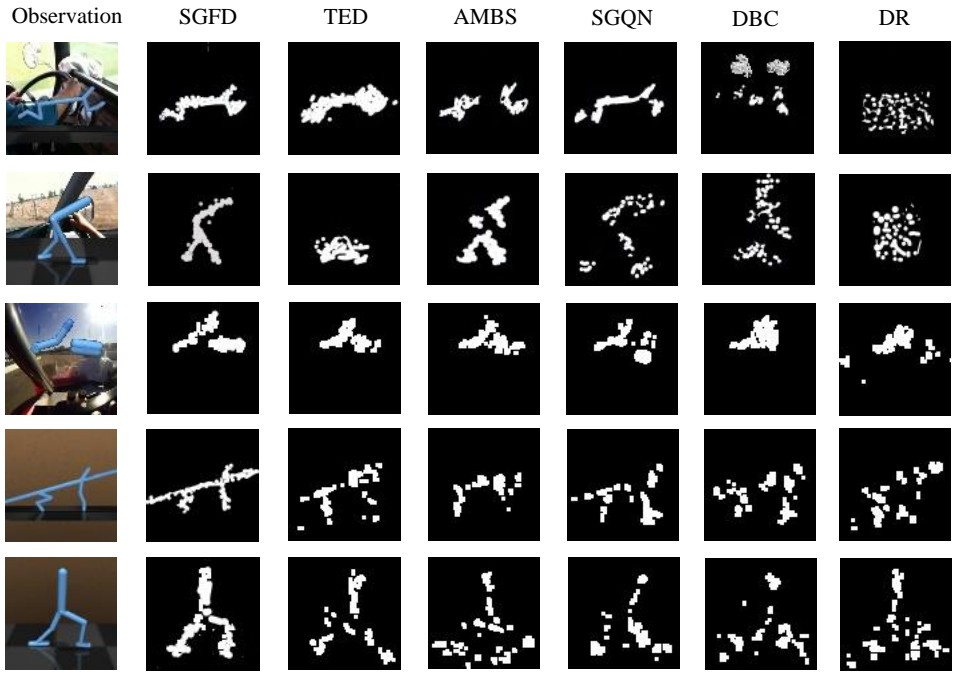

Figure 11: Visualization of saliency maps produced by the SGFD and other baselines when the robot with different backgrounds and robot parameters. The brighter the pixel is, the more contributions it makes to the decision.

## C.6 Evaluation on the Generalization of SGFD

**Evaluation on Changed Task-Irrelevant Features.** We report the training curves of SGFD in environments with different task-irrelevant features, where the end points of the curves correspond to the results in Table 1. As illustrated in Figure 13, our model achieved superior performance compared to other baselines in previously unseen background environments. As the task becomes difficult (Figure 13 (b) and Figure 13 (d)), the advantage of SGFD achieves more significance. The results demonstrate that the decorrelation is suitable for RL tasks with different task-irrelevant features, and SGFD exhibits strong generalization capabilities to new values of task-irrelevant features.

**Evaluation on Changed Task-Relevant Features.** We also report the training curves of SGFD in environments with different task-relevant features, where the end points of the curves correspond to the results in Table 2. As shown in Figure 14, the reward curve exhibited a consistent positive correlation between the number of training steps and the average reward in the testing environment under both interpolation and extrapolation setups. Generally speaking, the extrapolation setting is more challenging for generalization because it requires the model to fully understand how changed features affect optimal decisions. Benefiting from decorrelation, our model achieved superior performance compared to other baselines, where the gaps were more significant under the extrapolation setting, as illustrated in Figure 14 (f). Our results demonstrate that the decorrelation facilitates our model's understanding of the true associations between changed features and decisions, leading to strong generalization on new values of task-relevant features.

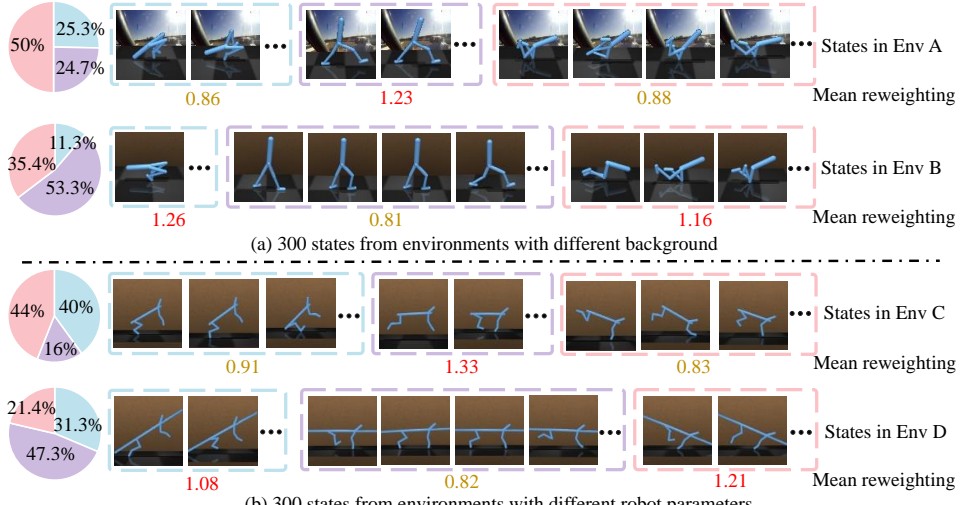

Figure 12: Visualization of weighted samples obtained from different environments with background noises and robot parameters. States that belong to the same dotted box with a specific color are considered semantically similar. SGFD balances the weights of samples to eliminate the correlations between changed features and other features. As shown in (a), the walker agents in Env A and Env B have different backgrounds, entangled with different distributions of the body posture. SGFD reduces the weighting of states with backgrounds and enhances the weighting of states without backgrounds, effectively neutralizing the correlation between background and body posture. A comparable result was observed in (b), where the reweighting process aimed to equalize the number of states in each group across environments with different body lengths.

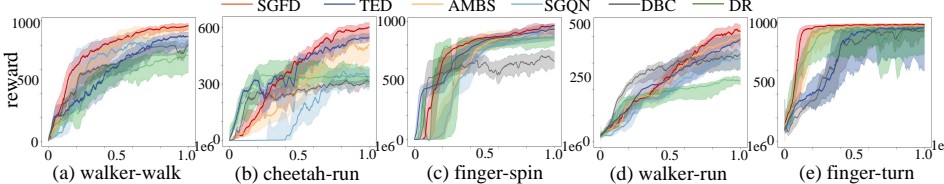

Figure 13: Generalization to changed task-irrelevant features. The x-axis denotes the number of training steps taken in training environments, while the y-axis indicates the average reward in the testing environment under different backgrounds. The reward curve demonstrates that our SGFD model generalizes well to new values of task-irrelevant features.

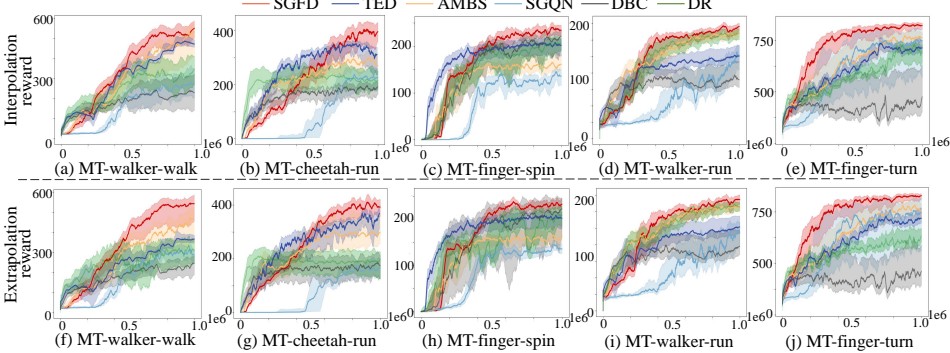

Figure 14: Generalization to changed task-relevant features. The x-axis denotes the number of training steps taken in training environments, while the y-axis represents the average reward in the testing environment with different robot parameters. We consider two setups for evaluation: an interpolation setup ((a)-(e)) and an extrapolation setup ((f)-(j)), where the changes in the task-relevant features are interpolations and extrapolations between the changes in the training environments, respectively. The reward curve shows that our SGFD generalizes well to the new values of task-relevant features.

