# OpenReview forum: "Learning Generalizable Agents via Saliency-guided Features Decorrelation"
_NeurIPS.cc/2023/Conference — NeurIPS 2023 spotlight_

### Official Review · Reviewer_NdCS · 2023-06-30

**Soundness:** 4 excellent
**Presentation:** 3 good
**Contribution:** 3 good
**Rating:** 7
**Confidence:** 3

**Summary:**

This paper introduces an original approach named Stochastic Gradient Feature Decorrelation (SGFD), which aims to amplify the generalization capabilities of reinforcement learning (RL) agents across various environmental variations. These variations can encompass task-irrelevant visual attributes such as backgrounds, as well as task-related factors like physical configurations. The authors propose achieving this through a decorrelation of features, executed via a resampling technique intended to minimize the Frobenius norm of the cross-covariance matrix, derived from Random Fourier features. Recognizing the inherent challenges of complete decorrelation, the authors shift focus towards effectively decorrelating the most variable features. This is facilitated by leveraging the saliencies from an environment classification model, which, under ideal circumstances, makes its decisions based on distinct features that are not common to different environments.

**Strengths:**

The paper is commendably well-written and coherent, effectively explaining complex ideas in an accessible manner. The authors demonstrate a strong theoretical grounding, with well-motivated intuitions supporting their methodology. Their saliency-guided optimization is an interesting approach, backed by a robust ablation study. SGFD successfully tackles generalization issues relating to both task-irrelevant and task-relevant features, demonstrating a broad scope of applicability. The proposed method do present noticeable enhancements in generalization performance, more so in the case of task-relevant features.


**Weaknesses:**

Despite the paper's strengths, there are some areas where it could be improved. Firstly, the methodology requires several environments with variations to train their environment classifier, introducing an element of manual supervision into the learning process, which may not be ideal in all scenarios. Furthermore, the full algorithm can be challenging to comprehend without first referring to Appendix A, suggesting that the main body of the text might benefit from additional clarification. It might also be beneficial to introduce the general objective - namely, the reweighting of the batch sampled from the buffer - earlier in the paper to give readers a clearer understanding of the process. Lastly, while the authors promote their method as an improvement over Soft Actor-Critic (SAC), the achieved results actually utilize the same updates for the encoder as employed by Adaptive Meta-learner of Behavioral Similarities (AMBS). This could lead to misunderstandings, as readers may infer that the saliency-guided resampling alone yields these performances. It would therefore be beneficial for clarity and fairness if the authors explicitly acknowledged this in the experimental section.


**Questions:**

* Could the authors elaborate on the inference procedure employed during testing? This could help elucidate the practical applicability of the methodology.
* Does the encoder also benefit from the gradients produced through resampling? Understanding this aspect could contribute to a more comprehensive understanding of the process.
* Given the necessity to decorrelate features that vary across environments from those that remain constant — and assuming that the variable features have a higher saliency — it appears that the $p\left(\mathbf{Z}_i\right) p\left(\mathbf{Z}_j\right)$ in equation 7 tends to assign more importance to tuples of varying features. Would it potentially be more efficient to replace it with $|p\left(\mathbf{Z}_i\right) - p\left(\mathbf{Z}_j\right)|$to accentuate the decorrelation between the varying and consistent features?


**Limitations:**

While the authors have reasonably addressed the methodological limitations of their approach, they assume no potential negative societal impacts stemming from their method.

---

> ### Author Rebuttal · Authors · 2023-08-08
>
> We are particularly encouraged that the Reviewer NdCs finds our method novel and effective.
>
> **Reply to the weakness**
>
> >**W1. Firstly, the methodology requires several environments with variations to train their environment classifier, introducing an element of manual supervision into the learning process, which may not be ideal in all scenarios.**
>
> We agree with the point of view. From a causal perspective, invariance needs to sum up from changing data. It is an intuitive way to train general policies from different environments. In the experiments, we set up four training environments for each task, which puts the cost in an acceptable range.
>
> >**W2. Furthermore, the full algorithm can be challenging to comprehend without first referring to Appendix A, suggesting that the main body of the text might benefit from additional clarification. It might also be beneficial to introduce the general objective - namely, the reweighting of the batch sampled from the buffer - earlier in the paper to give readers a clearer understanding of the process.**
>
> According to the suggestion, we split the original Figure 2 into two figures in the global response PDF. We clearly illustrate the motivation for feature decorrelation in Figure 1, visualizing the difference before and after sample reweighting. In Figure 2, we explicitly indicate that the model aims to reweight a batch of samples from the replay buffer. Based on the revised Figure 2, we give a more precise technical flow description.
>
> **Figure 2: the architecture of SGFD.** Essentially, our SGFD aims to reduce correlations in image features by reweighting samples. This involves five steps: (1) We fetch a sample batch from the replay buffer, which may come from multiple environments with different backgrounds or robot configurations. (2) The image in the sample is compressed by an encoder into latent features. (3) Then, we augment these features using multiple Random Fourier Functions to capture nonlinear correlations. (4) Concurrently, we train a classifier and apply saliency maps to detect features that shift across environments. (5) Finally, SGFD reweights samples to eliminate the correlations between identified features and other features.
>
> >**W3. Lastly, while the authors promote their method as an improvement over SAC, the achieved results actually utilize the same updates for the encoder as employed by Adaptive Meta-learner of Behavioral Similarities (AMBS). This could lead to misunderstandings, as readers may infer that the saliency-guided resampling alone yields these performances. It would therefore be beneficial for clarity and fairness if the authors explicitly acknowledged this in the experimental section.**
>
> Thanks for the suggestion; we will explicitly describe the details of the encoder in the experimental section, not just in the appendix.
>
> **Reply to questions**
> >**Q1. Could the authors elaborate on the inference procedure employed during testing?**
>
> Our method consists of four neural network models: an actor model (policy), a critic model, an encoder, and a classifier model. Among these four models, the critic model is used to improve the actor's performance during the training phase. At the same time, the classifier is used to assist the sample reweighting process. During testing, we deployed the encoder and actor model to the new environment. During inference, the encoder compresses the observed image into a latent representation, and the actor model predicts actions until the environment is terminated or the task is completed. This process will not involve the calculation of the classifier and the critic model.
>
> >**Q2. Does the encoder also benefit from the gradients produced through resampling?**
>
> The role of the encoder is to compress high-dimensional images into compact representations, which are the basis of other models. In our experiments, the encoder is only updated along with the critic model, which is not affected by sample reweighting. The sample reweighting is only applied to the actor model to avoid confusing which part of the model brings the final result improvement. We will clarify this in the final version to make it easier for readers to understand the process.
>
> >**Q3. Would it potentially be more efficient to replace $p(\mathbf{Z}_i)p(\mathbf{Z}_j)$ with $|p(\mathbf{Z}_i)-p(\mathbf{Z}_j)|$ to accentuate the decorrelation between the varying and consistent features?**
>
> Intuitively, $|p(\mathbf{Z}_i)-p(\mathbf{Z}_j)|$ can identify pairs of varying features and invariant features with high probability. According to your suggestion, we conducted the related experiment and compared it with Equation (7). Due to the time limit, we tested several scenarios with task-irrelevant situations.
>
> |  | walker-walk | cheetah-run | finger-spin | walker-run | finger-turn |
> |-------|-------|-------|-------|-------|-------|
> | $p(\mathbf{Z}_i)p(\mathbf{Z}_j)$ | $\mathbf{959.1 \pm 26.3}$ | 599.6 $\pm$ 47.2 | 965.7 $\pm$ 45.9 | $\mathbf{420.7 \pm 39.2}$ | $\mathbf{984.3 \pm 11.5}$ |
> | $\mathbf{abs}(p(\mathbf{Z}_i)-p(\mathbf{Z}_j))$ | 951.2 $\pm$ 27.2 | $\mathbf{605.6 \pm 46.6}$ | $\mathbf{975.5 \pm 39.8}$ | 406.3 $\pm$ 46.1 | 974.2 $\pm$ 13.5 |
>
> From the preliminary results, $|(p(\mathbf{Z}_i)-p(\mathbf{Z}_j)|$ achieved a comparable performance as $p(\mathbf{Z}_i)p(\mathbf{Z}_j)$. Two points may cause the fluctuation in performance: (1) Due to the data limit, sample reweighting only achieves approximate perfect decorrelation. (2) The classifier's error may affect the recognition of changed features. Overall, $|p(\mathbf{Z}_i)-p(\mathbf{Z}_j))|$ is a promising approach and will complement experiments in task-relevant situations. Supplementary experiments will be updated in the latest version.

---

> > ### Comment · Reviewer_NdCS · 2023-08-11
> >
> > I appreciate the authors' responses that clarified my questions. After reviewing their explanations, I maintain my opinion that this paper possesses the necessary qualities for acceptance.

---

### Official Review · Reviewer_SPDk · 2023-07-07

**Soundness:** 2 fair
**Presentation:** 3 good
**Contribution:** 2 fair
**Rating:** 7
**Confidence:** 2

**Summary:**

In visual-based Reinforcement Learning (RL), agents often struggle with generalization to environmental variations that were not observed during training that have changed task-irrelevant features and changed task-relevant features.
To achieve generalization in environmental variations, The authors popose a sample reweighting method for RL tasks that encourages the agent to understand the impact of changed features on its decisions, called Saliency-Guided Features Decorrelation (SGFD).  The authors demonstrated that SGFD significantly outperforms state-of-the-arts in handling changed task-relevant features.

**Strengths:**

This paper is well organized and described. Details of the proposed method are well described with supplementary materials. The performance advantages are not huge, but the authors have detailed experiments and analysis of changed feartures that support their claims.

**Weaknesses:**

The proposed method use the saliency to calculate and discriminate changed features under environmental changes.
Their experimental results and analysis include only one of task-irrelevant feature cases or task-relevant feature cases.
The proposed method is dependent on the saliency of the learned policy. Additional analysis of policy dependency would be helpful to support the authos' claims.

**Questions:**

Additional experiments and analysis of mixture of changes of task-irrelevant feature cases and task-releeant feature cases would be included.
Additional analysis of policy dependency would be helpful to support the authos claims.

**Limitations:**

The authors describe limitations, but more insights about limitations in real situations that include mixture of changes of task-irrelevant feature cases and task-relevant feature cases would be helpful.

---

> ### Author Rebuttal · Authors · 2023-08-06
>
> We are encouraged that Reviewer SPDk finds our method novel and the idea well-grounded.
>
> **Reply to the weakness**
> >**W1. The proposed method use the saliency to calculate and discriminate changed features under environmental changes. Their experimental results and analysis include only one of task-irrelevant feature cases or task-relevant feature cases.**
>
> Firstly, we wish to clarify that variations in task-relevant or task-irrelevant features are widespread in the real world. Therefore, many works focus on one of these situations and propose novel ideas [1-4]. Second, our work improves generalization in both cases and achieves comparable performance to other algorithms in their focus scenarios. Lastly, as suggested, we evaluate the most challenging environment where both task-relevant and task-irrelevant features are changed. Specifically, in the MT-cheetah-run task, both torso length and background noise varied during testing. Similarly, in the MT-finger-turn task, the robot arm's length and background noise changed. Neither the changed background nor the robot's configuration was present in the training environments.
>
> | Tasks | SGFD | TED | AMBS | SGQN | DBC | DR |
> |-------|-------|-------|-------|-------|-------|-------|
> | Mix-cheetah-run | $\mathbf{378.6 \pm 36.2}$ | 308.6 $\pm$ 47.6 | 265.6 $\pm$ 37.4 |  161.2 $\pm$ 47.5 |  171.2 $\pm$ 36.2 |  189.4 $\pm$ 46.3 |
> | Mix-finger-turn | $\mathbf{899.3 \pm 29.5}$ | 726.2 $\pm$ 52.2 | 802.5 $\pm$ 53.1 | 632.9 $\pm$ 52.8 |  321.3 $\pm$ 61.6 |  605.6 $\pm$ 57.8 |
>
> From the results, the advantage of feature decorrelation is further amplified and outperforms the baseline by 28% when the changes in both cases co-occur. This is because our method can handle the generalization problem of task-relevant and task-irrelevant cases, while previous work usually only focuses on one. In addition, feature decorrelation encourages the model to recover the true associations between the changed features and the optimal behavior and thus be able to adjust actions in unseen environments correctly. We will add this experiment to the latest version.
>
> >**W2. The proposed method is dependent on the saliency of the learned policy. Additional analysis of policy dependency would be helpful to support the authors' claims.**
>
> Based on the reviewer's comment on the weakness, we speculate that the "policy dependency" mentioned here means that the policy's generalization depends on the accuracy of the saliency map. In the paper, we performed two types of experiments directly related to the saliency map.
>
> The first class experiments are Figures 4 and 6 in the main text, which test the performance of our method without the assistance of saliency maps. As shown in Figure 4 (b), without the guidance of the saliency map, the decorrelation ability of the model is obviously limited, and this result also corresponds to the generalization of Figure 6 (b).
>
> Another type of experiment is Figure 9 in the appendix, which focuses on testing the accuracy of the classifier. From the results, the classifier achieves an accuracy of over 0.9 within 1e5 steps in each task, which demonstrates that saliency maps can accurately identify features that shift across environments.
>
> **Reply to questions**
> >**Q1. Additional experiments and analysis of mixture of changes of task-irrelevant feature cases and task-relevant feature cases would be included.**
>
> According to the suggestions, we evaluate our method on the mixture of changes of task-irrelevant feature cases and task-relevant feature cases. The experiment results can be find in the response of W1.
>
> >**Q2. Additional analysis of policy dependency would be helpful to support the authors claims.**
>
> Similarly, we discussed the "policy dependency" in our response to W2.
>
> **References**
> [1] Temporal disentanglement of representations for improved generalisation in reinforcement learning. ICLR 2023.
> [2] Look where you look! saliency-guided q-networks for generalization in visual reinforcement learning. NeurIPS 2022.
> [3] Learning generalizable representations for reinforcement learning via adaptive meta-learner of behavioral similarities. ICLR 2022.
> [4] Learning robust state abstractions for hidden-parameter block mdps. ICLR 2021.

---

> > ### Author Response · Authors · 2023-08-20
> > **Follow-up to Reviewer SPDk**
> >
> > Dear Reviewer SPDk,
> >
> > We value your positive feedbacks and constructive suggestions for our paper and sincerely appreciate your effort in reviewing it. As the end of the discussion is approaching, we kindly request your consideration regarding the possibility of raising the score. We thank the reviewer's effort in the review of our paper. We hope we have effectively addressed all the concerns raised. Should there be any remaining concerns, we stand ready to offer additional clarifications.
> >
> > Thank you again for your dedicated review and invaluable insights.
> >
> > Kind regards,
> >
> > Paper1844 Authors

---

> > > ### Comment · Reviewer_SPDk · 2023-08-21
> > >
> > > Thank you for the responses. I have read the reviews of the other reviewers and the authors' responses.  The authors' responses clarified my questions. Also, my concerns about dependency on the classifier part is well described in the appendix. Finally I have raised my score.

---

### Official Review · Reviewer_jEbK · 2023-07-10

**Soundness:** 3 good
**Presentation:** 3 good
**Contribution:** 3 good
**Rating:** 7
**Confidence:** 4

**Summary:**

This paper deals with a novel sample re-weighting method designed to enhance generalization in visual reinforcement learning tasks across environments with unseen task-irrelevant and task-relevant features. SGFD is composed of two core components: RFF and a saliency-guided model. The paper puts down results comparison with standard datasets to establish the claim. This is a very important problem w.r.t. many downstream tasks of agents where main reason for failures is due to lack of generalization in learning.

**Strengths:**

Code and supplementary material content are good.
The paper is well written with example in intro.
Appendix A,B are the crux of this paper in terms of contributions.

**Weaknesses:**

Related work needs an overhaul in terms of gaps and only putting exactly related prior art.
Contribution section please re-write and make it crisp to two main claims.
Apart from Pearson 218, other correlation checks could be put into function.


**Questions:**

Fig. 2 weighted samples in figure, why those weights and what value it adds?
Not clear why HSIC was discussed in first place, line 142.
Is not Figure 3 too perfect wrt other approaches? Why so? I will also like to see worst results in supplementary to see variance.
Table 2 - in term of metric results there exists 2 clusters (walker-walk, finger-spin,finger-turn) and (cheetah-run, walker-run). Is the are any analysis for this?

**Limitations:**

The work is limited to the specific datasets tried on in terms of training, otherwise adaptable.

---

> ### Author Rebuttal · Authors · 2023-08-08
>
> We are particularly encouraged that the Reviewer jEbK finds our method novel and effective.
>
>
> **Reply to the weakness**
> >**W1. Related work needs an overhaul in terms of gaps.**
>
> Building on the work discussed in Section 2, we further complement related work that generalizes to novel environment configurations. These works are typically studied with procedurally generated environments [1][2]. Some approaches to this problem leverage techniques from supervised learning, such as regularization, curriculum strategies, hyper-parameter tuning, and using self-supervised objectives [3-5]. Recurrent Independent Mechanisms leverage modules to learn a state function improved out-of-distribution generalization [6]. Feature-Attending Recurrent Modules showed that a modified attention mechanism led to strong generalization improvements with RL [7]. Our approach differs from these approaches in that we make no assumptions about the structure of task rewards or states, exploiting feature decorrelation to improve the generalization.
>
> >**W2. Contribution section please re-write and make it crisp to two main claims.**
>
> According to the suggestion, we revised the contribution part and made it crisp to two main claims. Summary of Contributions: (1) We propose the SGFD model that utilizes a sample reweighting method to improve generalization in visual RL tasks, covering both task-relevant and task-irrelevant situations. (2) Experimental results demonstrate that SGFD generalizes well on a wide range of novel environments and significantly outperforms state-of-the-art methods in handling task-relevant variations.
>
> >**W3. Apart from Pearson, other correlation checks could be put.**
>
> We additionally introduce Spearman's rank correlation coefficient to test our method. Since Spearman's rank correlation coefficient cannot be directly applied in the case of weighted data, we first randomly select 128 samples based on the distribution of weighted values and calculate the standard Spearman's rank correlation coefficient based on these 128 samples. To save space, we take Figure 7 in the appendix as an example and calculate the correlation between the changed features ($s_8-s_{10}$) with other features ($s_1-s_7$). Similar to the Pearson correlation coefficient, we control the value range between 0 and 1, where 1 means complete correlation and 0 means independent of each other.
>
> |Unweighted samples|$s_1$|$s_2$|$s_3$|$s_4$|$s_5$|$s_6$|$s_7$|
> |-|-|-|-|-|-|-|-|
> |$s_8$|0.81|0.72|0.78 |0.79|0.74|0.74|0.92|
> |$s_9$|0.81|0.82|0.89 |0.78|0.86|0.82|0.84|
> |$s_{10}$|0.79|0.75|0.80|0.91|0.87|0.89|0.74|
>
> |Weighted samples|$s_1$|$s_2$|$s_3$|$s_4$|$s_5$|$s_6$|$s_7$|
> |-|-|-|-|-|-|-|-|
> |$s_8$|0.43|0.16|0.31|0.24|0.35|0.14|0.20|
> |$s_9$|0.28|0.27|0.31|0.18|0.28|0.24|0.39|
> |$s_{10}$|0.28|0.32|0.31|0.36|0.31|0.38|0.24|
>
> From the results, $s_8-s_{10}$ and $s_1-s_7$ show a strong correlation in the unweighted data, which is significantly reduced after reweighting.
>
>
> **Reply to questions**
> >**Q1. Fig. 2 weighted samples in figure, why those weights and what value it adds?**
>
> The weights in Figure 2 could eliminate the correlation of features in the image, which is beneficial to improve the generalization. For ease of understanding, we discuss the motivation for feature decorrelation in Figure 1 in the general response PDF file. Consider a classifier trained on images labeled 'cat': three with sofas and one with grass in the background. The model may falsely link sofas to cats due to frequent pairing. To mitigate this issue, we can reweight the images to decorrelate the features between backgrounds and animals. In RL, features like varied background noise can correlate with the robot's state, as in Figures 5 (main text) and 11 (Appendix). Such correlations may lead agents to form spurious associations, altering optimal actions with background shifts.
>
> >**Q2. Not clear why HSIC was discussed in first place, line 142.**
>
> The HSIC is the theoretical motivation behind our approach. Therefore, we introduce HSIC, leading to our correlation evaluation method based on the Random Fourier Function.
>
> >**Q2. Is not Figure 3 too perfect wrt other approaches? Why so?**
>
> Figure 3 does not represent that our method achieves perfect results on every task. Specifically, each polygon vertex denotes a task, and we normalize all algorithms' performances against the state-of-the-art result. Figure 3 is a visualization of Table 1 and Table 2, so the variance of each task can be seen in Table 1 and Table 2. In addition, we also record the learning curve, and the shaded part of the curve also shows the variance of the performance, as shown in Fig. 12 and Fig. 13 in the appendix.
>
> >**Q3. Table 2 - there exists 2 clusters (walker-walk, finger-spin,finger-turn) and (cheetah-run, walker-run). Is the are any analysis for this?**
>
> Based on the description, the reviewer may be talking about the results in Table 1. Each task in Table 1 is derived from DeepMind Control Suite [8], which is a widely used benchmark. In this benchmark, each task has an independent reward setting, e.g., 600 is a bad score in finger-spin, but it is close to the optimal solution in cheetah-run. Overall, our method achieves decent results on every task. Therefore, two clusters are observed because they have similar maximum reward settings.
>
>
> **References**
> [1] Leveraging procedural generation to benchmark reinforcement learning. ICML 2020.
> [2] MiniHack the Planet: A Sandbox for Open-Ended Reinforcement Learning Research. NeurIPS 2021.
> [3] Reinforcement learning with augmented data. NeurIPS 2020.
> [4] Generalization in reinforcement learning with selective noise injection and information bottleneck. NeurIPS 2019.
> [5] Procedural generalization by planning with self-supervised world models. ICLR 2022.
> [6] Recurrent independent mechanisms. 2019.
> [7] Feature-attending recurrent modules for generalization in reinforcement learning. ICLR 2022.
> [8] Deepmind control suite.2018.

---

> > ### Comment · Reviewer_jEbK · 2023-08-18
> > **Intermediate comments**
> >
> > Thanks for the clarifications. I am sticking with my initial score while other reviewers are updating.

---

### Official Review · Reviewer_NDX8 · 2023-07-24

**Soundness:** 2 fair
**Presentation:** 2 fair
**Contribution:** 2 fair
**Rating:** 5
**Confidence:** 4

**Summary:**

This paper propose SGFD, an novel approach aimed at improving the generalization ability of distinguishing task-irrelevant and task-relevant situations. SGFD leverages two core techniques: Random Fourier Functions (RFF) and the saliency map, to estimate the complex non-linear correlations in high-dimensional images, and identify the changed features to achieve decorrelation. Furthermore, it conducts experiments on two benchmarks to exhibit its improved generalization ability.

**Strengths:**

1. Its proposed method has the capability to address problems in two distinct settings.

2. I think utilizing saliency map to further achieve decorrelation is an interesting idea.

**Weaknesses:**

1. Figure 2 is confusing. This figure is meant to illustrate the core content of the paper, its unclear presentation undermines the overall comprehensibility. For instance, it would be clearer if the authors had depicted only one type of environment. The inclusion of two types of environments creates ambiguity as to whether the training was performed in one or both environments.

2. This paper lacks a clear logical flow in the writing, particularly when it comes to the technical details.

3. The performance of the proposed SGFD method does not seem to significantly surpass that of other algorithms. The overlap within the standard deviation intervals between SGFD and other algorithms suggests that the advantages of SGFD are not as pronounced as might be expected.

**Questions:**

1. Can you visualize the examples of training enviroments and test environments?  I don't know the gap between these two environments and how difficult to generalize to the novel scenes.

2. Can you illustrate the Figure 4 to me? I think you need to decorrelate the association between features and the policy, not among different features? But the Figure 4 seems to decorrelate the features.

3. Why the calculated $w*$ only apply to actor loss? Why not critic?

4. Is it possible to apply SGFD on the environments which contains both task-relevant and task-irrelevant features?

If you can address my questions, I would consider increasing the score.

**Limitations:**

Mentioned in weakness and questions.

---

> ### Author Rebuttal · Authors · 2023-08-06
>
>
> Thanks for the reviewer's positive appraisal, insightful comment, and criticism of our paper.
>
> **Reply to the weakness**
> >**W1. Figure 2 is confusing.**
>
> We split the original Figure 2 into two figures in the global response PDF: Figure 1 discusses the motivation for feature decorrelation, while Figure 2 details the technique flow.
>
> **Figure 1: the motivation for feature decorrelation.** Consider a classifier trained on images labeled 'cat': three with sofas and one with grass in the background. The model may falsely link sofas to cats due to frequent pairing. To mitigate this issue, we can reweight the images to decorrelate the features between backgrounds and animals. In RL, features like varied background noise can correlate with the robot's state, as in Figures 5 (main text) and 11 (Appendix). Such correlations may lead agents to form spurious associations, altering optimal actions with background shifts.
>
> **Figure 2: the architecture of SGFD.** Essentially, our SGFD aims to reduce correlations in image features by reweighting samples. This involves five steps: (1) We fetch a sample batch from the replay buffer, which may come from multiple environments with different backgrounds or robot configurations. (2) The image in the sample is compressed by an encoder into latent features. (3) Then, we augment these features using multiple Random Fourier Functions to capture nonlinear correlations. (4) Concurrently, we train a classifier and apply saliency maps to detect features that shift across environments. (5) Finally, SGFD reweights samples to eliminate the correlations between identified features and other features.
> >**W2. This paper lacks a clear logical flow in the writing, particularly when it comes to the technical details.**
>
> Our writing logic is as follows. In the Introduction, we first discuss the generalization when task-relevant and task-irrelevant features are shifted and point out that it can be achieved by feature decorrelation. Then, the Preliminary Section indicates that the learned encoder obtains the image features. The Method Section starts with using the Random Fourier Function for assessing feature correlation and elaborates on decorrelation through sample reweighting. It then delves into the challenge of decorrelating all feature pairs, introducing the saliency map to focus on shifted features. Finally, Our experiments in Sections 5.2 to 5.4 assess generalization, feature decorrelation, and conduct ablation studies. This logic flow will be clearer in the updated version with the revised Figure 2.
> >**W3. SGFD does not seem to significantly surpass other algorithms.**
>
> The advantages of our SGFD method are reflected in two aspects. Firstly, it improves generalization for both task-relevant and task-irrelevant cases, while previous work usually only focuses on one. Secondly, it shows clear superiority in task-relevant cases, outperforming the state-of-the-art method by 23% on average, as shown in Table 2. This is because feature decorrelation encourages the model to form true associations with the features and thus be able to adjust actions in unseen environments correctly.
>
> **Reply to questions**
> >**Q1. Can you visualize the training and test environments?**
>
> We visualize the environments in the "global" response PDF file. As shown in Figure 3, the model is trained in 4 environments with different task-irrelevant features or task-relevant features. During testing, the model is deployed in unseen environments without extra training. Differential settings between training and testing follow the baseline algorithms.
> >**Q2. Can you illustrate the Figure 4? Why eliminate the correlation between features？**
>
> **Illustration Figure 4:** Figure 4 depicts a 10x10 heatmap, visualizing the correlations between ten features ($s_1-s_{10}$) in a robotic arm grasping task. To compute the correlation of weighted data, we fetch a sample batch from the weighted distribution and then calculate the Pearson correlation coefficient. The model could access true states ($s_1-s_{10}$) with physical meaning. Notably, $s_{10}$ denotes the object's size, which varies across environments. For optimal RL model generalization to objects of unobserved sizes, it's crucial to assess correlations between $s_{10}$ and $s_1-s_9$. As highlighted in Figure 4, our method successfully reduces this correlation.
>
> **Why decorrelation:** Due to the word limit, please refer to the motivation for feature decorrelation in the W1 response.
>
> **Decorrelate associations between features and policies:** In fact, feature decorrelation leads the policy to decorrelate with some features. As the example in reply W1, the policy will eliminate the dependence of the background on decision-making when the semantics of images are independent of the background.
> >**Q3. Why the calculated $w$ only apply to actor? Why not critic?**
>
> With the actor loss under the $w^*$ effect, the model could recover the true associations between the changed features and the optimal actions. In the testing phase, we only deploy the policy network to the new environment without transferring the critic; thus, $w^*$ does not need to change the critic loss.
> >**Q4. Can you apply SGFD on both task-relevant and irrelevant features?**
>
> As suggested, we test the environment where both task-relevant and task-irrelevant features are changed. Specifically, in the MT-cheetah-run task, both torso length and background noise varied during testing. Similarly, in the MT-finger-turn task, the robot arm's length and background noise changed.
>
> |Tasks|SGFD|TED|AMBS|SGQN|DBC|DR|
> |-|-|-|-|-|-|-|
> |Mix-cheetah-run|$378.6±36.2$|308.6±47.6|265.6±37.4|161.2± 47.5|171.2±36.2|189.4±46.3|
> |Mix-finger-turn|$899.3±29.5$|726.2±52.2|802.5±53.1|632.9± 52.8|321.3±61.6|605.6±57.8|
>
> From the results, the advantage of feature decorrelation is further amplified and outperforms the baseline by 28% when the changes in both cases co-occur. We will add this experiment to the latest version.

---

> > ### Comment · Reviewer_NDX8 · 2023-08-15
> > **Reply to Authors**
> >
> > Thanks for your reply. I have several other questions. Regarding Q3, would the calculated weight be effective on any objective? Why must it only be placed on the actor? It feels to me that there is some disconnection between how to obtain $w$ and how to apply it to RL. Moreover, your answer to Q3 seems subjective; many generalization papers (e.g, SVEA, SGQN) have made modifications to the critic objective, and they are also very effective.
> >
> > Regarding Q4, the effect of the finger turn appears very good, but in fact, you have not shown any visualizations related to anything beyond the cheetah task, leaving the reader in the dark about how you modified your environment.

---

> > > ### Author Response · Authors · 2023-08-16
> > > **Reply to the Reviewer NDX8**
> > >
> > > We thank Reviewer NDX8 for further response to our rebuttal. We want to further clarify some of the questions as below.
> > >
> > > **Q1. Why must $w$ only be placed on the actor? Would the calculated weight be effective on any objective?**
> > >
> > > **Why only apply $w$ to the actor:** It is possible to apply $w$ to the critic. We apply $w$ to the actor for the following two reasons.
> > >
> > > - The core idea of sample reweighting is to eliminate the correlation between features and then facilitate the policy to distinguish the impact of each image feature on decision-making. Since the actor directly controls the training of the policy, incorporating $w$ into the actor's loss is an intuitive and efficient method. Although modifying the critic loss is incorporated into the algorithm in related works, the critic can only indirectly benefit the policy on $w$ in our method, which is demonstrated in experiments (see the next point).
> > > - We conduct ablation experiments to test generalization under four settings: no $w$, $w$ only on the actor, $w$ only on the critic, and $w$ on both the actor and the critic. As shown in Table 1, when $w$ acts on the actor, the generalization is significantly improved, while the gain realized by the critic is relatively modest. This is because the actor loss directly controls the learning of the policy so that $w$ applied to the critic alone cannot prevent the policy from establishing false connections with the background noise. In addition, we do not observe significant improvements when adding $w$ to both the actor and the critic, compared to adding $w$ only to the actor. This further demonstrates that actors with $w$ adequately mitigate spurious associations caused by correlations between features.
> > >
> > > **Table 1:** The generalization of the $w$ acting on different parts of the algorithm in unseen background noises.
> > > |  | both the actor and critic | only the actor | only the critic | no $w$ |
> > > |-------|-------|-------|-------|-------|
> > > | walker-walk | 956.3 | $\mathbf{959.1}$ | 915.2 | 906.7 |
> > > | cheetah-run | $\mathbf{606.5}$ | 599.6  | 521.6 | 517.7 |
> > > | finger-spin | 958.6 | $\mathbf{965.7}$ | 905.5 | 895.5 |
> > >
> > > **Would the calculated weight be effective on any objective?:** Although the SAC algorithm is the backbone of our method, the form of sample reweighting makes it possible to transfer to other RL algorithms. Based on the experimental results, we propose to apply $w$ to the loss directly associated with the policy. For the RL method without the policy networks, we will further explore the modification of $w$ in future work.
> > >
> > > **Q2. The visualizations of finger turn.**
> > >
> > > Due to the page limit of the global rebuttal, we only visualized the configuration of the cheetah. According to the official regulations of NeurIPS 2023, we provided an anonymous link in the AC comment to visualize the configuration of the three robots used in our experiments. After AC's confirmation of anonymity, you can see it in the comments at the top. Specifically, the settings of the Finger and walker are referred to [1], which modifies the Finger's length and walker's foot. For background noise, Finger and cheetah are similar to the walker we showed in the global rebuttal PDF (Figure 3 (a)). The visualization will be updated to the revised version.
> > >
> > > **Reference**
> > > [1] Learning robust state abstractions for hidden-parameter block mdps, ICLR 2021.
> > >
> > > -----
> > > We hope that our response can address your comments and we would appreciate it if you considered increasing your score.
> > >
> > > Paper1844 Authors

---

> > > > ### Comment · Reviewer_NDX8 · 2023-08-18
> > > > **Reply to Authors**
> > > >
> > > > Thanks for your further reply.
> > > > - SGQN is an algorithm employed in a different setting. Yet It is also proper to apply it on your setting. I wonder in your setting, is it also trained in a noisy environment and subsequently tested in unseen noisy scenarios?
> > > > - Could you show me the training curve? Is the training performance of SGFD better than other algorithms? Because, I think the level of generalization you have shown appears to be much simpler than DMCGB. If SGQN can achieve high scores in the training scenarios, it is unlikely to experience a significant performance drop.

---

> > > > > ### Author Response · Authors · 2023-08-18
> > > > > **Reply to Reviewer NDX8**
> > > > >
> > > > > We thank the reviewer for the response, and the following are our further clarifications.
> > > > >
> > > > > **Q1. I wonder in your setting, is it also trained in a noisy environment and subsequently tested in unseen noisy scenarios?**
> > > > >
> > > > > Testing in the scene with unseen background noise is one of our settings, i.e., task-irrelevant features. The background noise follows the setting of AMBS [1] in the comparison algorithm. However, we want to emphasize that our SGFD can not only handle background noise, but also generalize to scenarios with **unseen configurations of robots** (i.e., task-relevant features), while the latter is not covered by SGQN. Unlike background noise, the differences in robot configurations mean that the optimal action also changes [2]. Therefore, from the causal perspective, we propose feature decorrelation to ensure that the policy recovers the true associations between changed features and optimal behavior, whether the changed features are task-relevant or task-irrelevant.
> > > > >
> > > > > **Q2. Could you show me the training curve? Is the training performance of SGFD better than other algorithms?**
> > > > >
> > > > > Although, according to the rules of Neurips 2023, we cannot visualize the training curve in the comment, Figures 12 and 13 in the appendix show similar results. Figure 12 and Figure 13 visualize the learning curves under two settings of unseen background noises (task-irrelevant) and robot configurations (task-relevant), respectively. The x-axis represents the number of steps in the training environment, and **the y-axis represents the reward obtained in the test environment**.
> > > > >
> > > > > **The training curves in environments with background noises:** For the setting of background noise, the training curves of SGQN are mostly consistent with the curves in Figure 12. Aided by feature decorrelation, our method performs better than SGQN in noisy training environments.
> > > > >
> > > > > **The training curves in environments with different robot configurations:** Since SGQN does not cover task-relevant generalization scenarios, its test performance is worse than its training performance under the robot configuration change scenario. In contrast, our method can generalize well to different robot configurations and outperform SGQN as shown in Figure 13.
> > > > >
> > > > > **The level of generalization and the comparison to DMCGB:** We want to clarify again that SGFD is proposed to expand the types of generalization, i.e., from task-irrelevant features to task-relevant features. It is worth acknowledging that the DMCGB adds more disturbing noise to the test environment, and SGQN proposes an efficient framework. However, we provide an orthogonal approach that attempts to achieve more types of generalization, which is in a different direction than their exploration. We will update this part of the discussion to the revised version.
> > > > >
> > > > > **References**
> > > > > [1] Learning Generalizable Representations for Reinforcement Learning via Adaptive Meta-learner of Behavioral Similarities. ICLR 2022.
> > > > > [2] Temporal disentanglement of representations for improved generalisation in reinforcement learning. ICLR 2023.

---

> > > > > > ### Author Response · Authors · 2023-08-20
> > > > > > **Follow-up to Reviewer NDX8**
> > > > > >
> > > > > > Dear Reviewer NDX8,
> > > > > >
> > > > > > We would like to express our sincere gratitude to you for reviewing our paper and providing valuable feedback. Could we kindly know if the responses have addressed your concerns? If there are any further questions, we are happy to clarify. Thank you.
> > > > > >
> > > > > > Best,
> > > > > >
> > > > > > Paper1844 Authors

---

> > > > > > > ### Comment · Reviewer_NDX8 · 2023-08-20
> > > > > > > **Reply to Authors**
> > > > > > >
> > > > > > > Thanks for your response. I will raise my score. Since I still haven't seen the visualization results,  my main concern is that whether DMC is a suitable  benchmark for your specific two settings. Additionally, from the visualization images you provided, your generalization can be considered as in-distribution generalization. For this type of generalization, I believe that showing the training curve is essential.

---

### Official Review · Reviewer_tXZg · 2023-07-25

**Soundness:** 3 good
**Presentation:** 2 fair
**Contribution:** 3 good
**Rating:** 7
**Confidence:** 4

**Summary:**

This paper presents a method to weight experiences in reinforcement learning so that the state features they depict are maximally decorrelated. The assumption is that learning with decorrelated features leads to better generalization to unseen task instances because the policy does not confound the role of different features when making decisions. The idea is novel and interesting and the experimental evaluation includes several simulated embodied AI tasks and comparisons to other baselines.


**Strengths:**

- The presented method is novel. The ideas of the paper are well-grounded.
- The method seems mathematically sound. The appendix includes all relevant proofs.
- The experimental evaluation, including several baselines and different tasks, demonstrates the strengths of the algorithm
- The problem addressed in the paper is critical for the robot learning community: improving the generalization capabilities of trained policies.



**Weaknesses:**

- The writing could be improved. The text focuses on describing the steps but would benefit from explaining more the reasoning behind the design decisions.
- The assumptions on the problem structure are unclear. What type of relationship between features and actions and features-to-features should be present in the problem? What happens if features are not (cannot be) decorrelated?
- It is unclear what the features are. I can’t find that information either in the text or in the appendix, maybe I’m missing it. Please, clearly discuss what the features are and whey they come from. Include it in the main text, e.g., in the experimental setup description. Explain and discuss how they are selected/created. In these times of representation learning, especially for vision-based control, is this a strong limitation of the method?
- Fig. 2 is quite complex. What are the colors of the “enriched features” indicating? What are the “G”s indicating? This figure could serve as a good summary of the method but it needs to be improved for that and include a more complete caption.



**Questions:**

- Explain the features per task and where they come from. Discuss the applicability of the method if it requires pre-specifying optimal features. How would the method work with learned latent features? Could you include an evaluation on that?
- Can you include completely distracting features to evaluate the robustness of the solution?
- Please, discuss what would be necessary for this method to work directly with images as input. It feels strange to only go “the other way”: from features to images (saliency) in order to be able to check when they are co-ocurring, but there is no discussion about the normal direction: from images to features.
- How does the method scale with the number of features? And with the number of samples? What is the computational cost? Can you experiment with that and provide some evaluation?


**Limitations:**

- The limitations section is acceptable, but several significant limitations for the method to be truly impactful are not discussed (see my comments above). Please, include them.

---

> ### Author Rebuttal · Authors · 2023-08-06
>
> We are particularly encouraged that the reviewer finds our method novel and effective.
>
> **Reply to the weakness**
> >**W1. Explaining more the reasoning behind the design decisions.**
>
> As suggested, we introduce an example to explain why feature decorrelation can promote generalization, as shown in Figure 1 in the "global" response PDF file.
>
> **Figure 1: the motivation for feature decorrelation.** Consider a classifier trained on images labeled 'cat': three with sofas and one with grass in the background. The model may falsely link sofas to cats due to frequent pairing. To mitigate this issue, we can reweight the images to decorrelate the features between backgrounds and animals. In RL, features like varied background noise can correlate with the robot's state, as in Figures 5 (main text) and 11 (Appendix). Such correlations may lead agents to form spurious associations, altering optimal actions with background shifts.
>
> This example can be first proposed in the method section and then fluently lead to the motivation of feature decorrelation.
>
> >**W2. What type of relationship between features and actions and features-to-features should be present? What happens if features cannot be decorrelated?**
>
> **Features-to-actions and inter-feature relationships.**
> We aim to recover the true association between features and actions, and for inter-feature relationships, we target their decorrelation. In W1's example, sample reweighting decorrelate the robot's state from the background, facilitating the model to recognize that the true association between background features and actions is 0.
>
> **What happens if features cannot be decorrelated?** In Figure 6 (a), the generalization will be hurt if the features are not decorrelated. In practice, due to the limited data, the sample reweighting often yields approximately independent results, not completely independent, as shown in Figure 4.
>
> >**W3. What the features are and where they come from.**
>
> Our method takes images as inputs, extracting latent features via an encoder trained alongside RL models. In our method, we leverage saliency map to identify features that vary across environments. In the main text, latent features are first mentioned on line 124. The structure of the encoder is mentioned in line 478 of the appendix.
>
> >**W4. Fig. 2 is quite complex. What are the colors of the “enriched features” indicating? What are the “G”s indicating?**
>
> The term "enriched features" refers to latent features mapped by multiple RRFs to assess nonlinear correlation. The symbol "G," initially representing the robot's center of gravity, was removed to prevent confusion. This is clarified in the updated Figure 2 in the "global" response PDF. We've replaced "enriched features" with "RFF map" for clarity, referencing Equation (4). The architecture now aligns better with the method's two subsections.
>
> **Figure 2: the architecture of SGFD.** Essentially, our SGFD aims to reduce correlations in image features by reweighting samples. This involves five steps: (1) We fetch a sample batch from the replay buffer, which may come from multiple environments with different backgrounds or robot configurations. (2) The image in the sample is compressed by an encoder into latent features. (3) Then, we augment these features using multiple RRFs to capture nonlinear correlations. (4) Concurrently, we train a classifier and apply saliency maps to detect features that shift across environments. (5) Finally, SGFD reweights samples to eliminate the correlations between identified features and other features.
>
> **Reply to questions**
> >**Q1. How would the method work with learned latent features? Could you include an evaluation on that?**
>
> As mentioned in W3, our method processes images directly, with the encoder utilizing the AMBS [1] update method alongside RL models. We'll detail the experimental setup in the main text, not just the appendix.
>
> >**Q2. Can you include completely distracting features to evaluate the robustness of the solution?**
>
> In Table 1, test environments have all distracting features. Inputs are 3x84x84 RGB pixel values, perturbed by real photos. We visualized the training and test environments in Figure 3 in the "global" response PDF file.
>
> >**Q3. What would be necessary for this method to work directly with images as input.**
>
> Our method takes images as input and uses an encoder to transform pixels into latent features. This encoder serves both the classifier and RL models. In experiments, the gradients from saliency maps are only propagated to the latent features, not the initial image layer.
>
> >**Q4. How does the method scale with the number of features? And with the number of samples? What is the computational cost?**
>
> **Number of features** As the inputs are images, the number of features is assumed to be the output dimension of the encoder. When accessing the ground-truth state of the environment, we demonstrate the decorrelation ability of the model for different numbers of changed features in Figure 4 of the main text and Figure 7 of the Appendix.
>
> **Number of samples and computational cost** The computational cost grows linearly with batch size. Using an off-policy RL algorithm, the cost is O(N x B), with N as environment steps and B as sample batch size.
>
> For clarity on computational cost, we've timed training with and without SGFD for batch sizes 64, 128, and 256. The total number of steps is 1 million, tested on an NVIDIA RTX 3090 GPU and Intel Xeon Platinum 8255c CPU at 2.50 GHz.
>
> |batch size|64|128|256|
> |-|-|-|-|
> |SGFD|13 h|16 h|22 h|
> |no SGFD|12 h|15 h|20 h|
>
> The results show that SGFD indeed brings extra computational cost due to the sample reweighting for decorrelation. However, compared to the RL algorithm, SGFD brings significant generalization at a cost of less than 10%.
>
> **References**
> [1] Learning generalizable representations for reinforcement learning via adaptive meta-learner of behavioral similarities. ICLR 2022.

---

> > ### Comment · Reviewer_tXZg · 2023-08-13
> >
> > Thanks for the replies, they clarify some misconceptions. I have raised my score.

---

### Author Rebuttal · Authors · 2023-08-09

Dear Reviewers,

We are very grateful to the reviewers for their valuable suggestions, which further improved our work. We provide three visualizations about our motivation, technique, and environmental setting with a submitted 1-page pdf.

- Figure 1: The motivation for feature decorrelation used in our method.
- Figure 2: The technique flow of our method.
- Figure 3: The visualization of training and testing environments.

Thank you again for your careful review and helpful comments.

Kind regards,

Paper1844 Authors

---

### Decision · Program_Chairs · 2023-09-21

**Decision:**

Accept (spotlight)

**Comment:**

This paper proposes a method to improvement generalization in visual-based reinforcement learning by training the policy-network to ignore the spurious correlation of input features present in the training data. This is achieved by using gradient-based saliency to guide the decorrelation of features extracted from input images. The authors adequately addressed the concerns raised by reviewers and the overall consensus is that this paper meets the bar for acceptance.